



# What effect does VOC sampling time have on derived OH reactivity?

**H. Sonderfeld[1], I.R. White[1], I.C.A. Goodall[1], J.R. Hopkins[2], A.C. Lewis[2], R. Koppmann[3], P.S. Monks[1]**

[1]{Department of Chemistry, University of Leicester, Leicester, LE1 7RH, UK}

[2]{National Centre for Atmospheric Science, University of York, York, YO10 5DD, UK}

[3]{Institute for Atmospheric and Environmental Research, University of Wuppertal, 42119 Wuppertal, Germany}

Correspondence to: P.S. Monks (P.S.Monks@leicester.ac.uk)

**Abstract**

State of the art techniques allow for rapid measurements of total OH reactivity. Unknown sinks of OH and oxidation processes in the atmosphere have been attributed to what has been termed 'missing' OH reactivity. Often overlooked are the differences in timescales over which the diverse measurement techniques operate. Volatile organic compounds (VOC) acting as sinks of OH are often measured by gas chromatography (GC) methods which provide low frequency measurements on a timescale of hours, while sampling times are generally only a few minutes. Here, the effect of the sampling time and thus the contribution of unmeasured VOC variability on OH reactivity is investigated. Measurements of VOC mixing ratios by proton transfer reaction time-of-flight mass spectrometry (PTR-ToF-MS) conducted during two field campaigns (ClearfLo and PARADE) in an urban and a semi-rural environment were used to calculate OH reactivity. VOC were selected to represent variability for different compound classes. Data were averaged over different time intervals to simulate lower time resolutions and were then compared to the mean hourly OH reactivity. The results show deviations in the range of 1 to 25%. The observed impact of VOC variability is found to be greater for the semi-rural site.

The selected compounds were scaled by the contribution of their compound class to the total OH reactivity from VOC based on concurrent gas chromatography measurements conducted





during the ClearfLo campaign. Prior to being scaled, the variable signal of aromatic
compounds results in larger deviations in OH reactivity for short sampling intervals compared
to oxygenated VOC (OVOC). However, once scaled with their lower share during the
ClearfLo campaign this effect was reduced. No seasonal effect on the OH reactivity
distribution across different VOC was observed at the urban site.
**1   Introduction**
Atmospheric photochemistry produces a variety of radicals that exert a substantial influence
on the ultimate composition of the atmosphere. The OH radical is the main oxidant in the
atmosphere (Monks et al., 2009 and references therein). Its actual concentration being
determined by the balance between its sources and sinks. While in many cases OH sources are
well understood, it sinks are manifold and not completely characterised. OH reactivity is a
measure of the strength of the sinks for the OH radical. It can be derived from the reaction
rates of the reactants $k_{OH+X}$ and their concentrations [X] (Kovacs et al., 2003):
$$k_{OH} = \sum k_{OH+VOC_i}[VOC_i] + k_{OH+CO}[CO] + k_{OH+NO}[NO] + k_{OH+NO_2}[NO_2] + k_{OH+SO_2}[SO_2] + ... \quad (1)$$
In-situ measurements of OH reactivity have provided new insights into OH loss chemistry
and the oxidative ability of the atmosphere (Di Carlo et al., 2004;Edwards et al.,
2013;Hofzumahaus et al., 2009;Whalley et al., 2011;Yoshino et al., 2006). There are a
number of different techniques used for the direct measurement of OH reactivity. The total
OH loss rate measurement technique (TOHLM) was one of the first techniques applied for
determination of total OH reactivity based on a single measurement (Ingham et al., 2009;Ren
et al., 2003a;Shirley et al., 2006). TOHLM is based on the measurement of the decay of
artificially produced OH following the introduction of reactants into an ambient air sample
within a flow tube. By varying the distance between the OH injection point and the detector,
the reaction time changes and provides a series of relative decay rates (Kovacs and Brune,
2001;Kovacs et al., 2003). A similar approach is taken with the laser-induced pump and probe
technique, whereby decay in OH is detected by time-resolved laser-induced fluorescence
(Sadanaga et al., 2004). Another technique developed by Sinha et al. (2008) called
Comparative Reactivity Method (CRM) is based on the measurement of a single reactant
(most often pyrrole) which first reacts with OH under clean air conditions and then under
competitive conditions with ambient air. The reaction takes place in a glass vessel and is most





commonly probed by PTR-MS. Recently, Nölscher et al. (2012b) presented a GC-PID for the
detection of pyrrole for CRM.
These techniques enable comparison of directly measured OH reactivity to calculated OH
reactivity using equation (1) based on measurements of individual compounds. The difference
between the two, is being referred to as missing OH reactivity. Reasons for an under
prediction of OH reactivity maybe due to incomplete or inaccurate measurements of
individual compounds (Di Carlo et al., 2004;Kim et al., 2011;Kovacs and Brune, 2001).
Therefore, direct measurements of total OH reactivity can help to evaluate the completeness
of measured VOC budgets (Dolgorouky et al., 2012;Mao et al., 2009;Mogensen et al., 2011).
In urban environments good agreement between measured and calculated OH reactivity have
been found. For example, no significant missing OH reactivity was found in New York during
summer (Ren et al., 2003b) and for both Paris under clean marine air conditions (Dolgorouky
et al., 2012) and Tokyo (Yoshino et al., 2006) in the winter. Larger missing OH reactivity of
up to 30% was found for all other seasons in Tokyo by Yoshino et al. (2006), presumably
owing to secondary reaction products, including semi volatile oxygenated compounds, from
atmospheric oxidation of VOC. A similar amount of missing OH reactivity was reported by
Kovacs et al. (2003) for urban measurements in Nashville. They suggest that non measured
short lived VOC accounted for the missing reactivity. In Paris, a missing OH reactivity of up
to 75% was found for continentally influenced air, which is also attributed to highly oxidized
compounds from photochemical processes during transportation of these air masses
(Dolgorouky et al., 2012). Similar reasons were reported by Lou et al. (2010) to account for
missing OH reactivity measured in the highly populated Pearl River Delta.
Direct measurements of OH reactivity in rural areas generally tend to have larger missing OH
reactivity. Using PTR-MS and the CRM method in a boreal forest in Finland during August
2008, Sinha et al. (2010) reported missing OH reactivity of approximately 50%. This site was
revisited in 2010, when missing OH reactivity of 58% to 89% was recorded (Nölscher et al.,
2012a). Similar results in a mixed deciduous forest where obtained by Hansen et al. (2014)
who reported missing OH reactivity of 46% to 65%. Both studies concluded that unmeasured
oxidation products were missing from the OH reactivity calculation. In contrast to those
findings, Ren et al. (2006) found no significant missing OH reactivity on average during a
summertime campaign in a deciduous forest in New York in 2002. They attributed this to
differences in the composition of emitted biogenic VOC (BVOC). Rainforests are a large



sink for OH as they emit a huge amount of VOC. Measured OH reactivity in the rainforest of
Borneo during April 2008 yielded a missing OH reactivity of 70% compared to calculated
reactivity from measurements of single compounds (Edwards et al., 2013) and ~53%
compared to modelled reactivity (Whalley et al., 2011). Since isoprene makes up the biggest
contribution to OH reactivity the effect of oxidation products of isoprene were discussed
(Edwards et al., 2013;Whalley et al., 2011).
While different possible explanations for missing OH reactivity have been given, the wide
range in reported missing OH reactivity suggests that many reactants and processes remain
unknown or cannot be measured at present. Measurements of total non-methane organic
carbon in the West Los Angeles Basin (Chung et al., 2003) and results following the
application of a double-column (orthogonal) GC for urban air measurements (Lewis et al.,
2000) emphasize the large number of OH reactants that are not measured with standard field
equipment.
Measurements of non methane hydrocarbons (NMHC) used for calculation of OH reactivity
are often performed with GC (Lou et al., 2010;Sadanaga et al., 2005;Shirley et al., 2006) and
therefore the time resolution of the calculated OH reactivity is low due to sample run times up
to 90 min (Dolgorouky et al., 2012), when compared to measured total OH reactivity.
However, the sampling time during one GC cycle is shorter than the analysis time and thus,
any high temporal variability in measured OH reactivity is not easily captured when it is
derived from GC data (Nölscher et al., 2012a). When measured and calculated OH reactivity
are compared, high time resolution data are often averaged over intervals that correspond to
the GC cycle.
This work addresses the question of how temporal VOC concentration variability is reflected
with different sampling time resolutions. Furthermore, the effect of averaging VOC data on
calculated OH reactivity is discussed alongside how this may affect the amount of so called
`missing` OH reactivity.
Relatively high time resolved VOC data collected by PTR-ToF-MS are used to calculate OH
reactivity for selected compounds. Differing time resolutions are analysed to explore the
effects. Data from an urban winter campaign are compared to measurements from a semi-
rural summer campaign.



**2 Experimental section**
Two different sets of VOC mixing ratios measured with PTR-ToF-MS were used for analysis.
One was collected during the ClearfLo (Clean Air for London, www.clearflo.ac.uk)
(Bohnenstengel et al., 2015) winter campaign in 2012 at an urban background site in London,
UK. The second was taken during the PARADE (PArticles and RAdicals: Diel observations
of the impact of urban and biogenic Emissions, http://parade2011.mpich.de/) campaign in late
summer 2011 at a semi-rural site located in the Taunus ridge, Germany.
**2.1 Field data**
**ClearfLo**. A PTR-ToF-MS (Series I; Kore Technology Ltd., UK) (Barber et al.,
2012;Thalman et al., 2014) was deployed at Sion Manning School (51°31'15" N, 0°12'51"
W) nearby the North Kensington urban background station in London during the intensive
observation periods of the ClearfLo project in 2012. A general overview of the ClearfLo
project and the measurement site is given in (Bohnenstengel et al., 2015). For background
measurements a hydrocarbon trap was employed. Calibration measurements were performed
before the campaign. For the calibration of toluene and xylene a permeation tube was used
and calibration of acetone was done with Tedlar bags containing different dilutions of an
acetone standard. The stability of the instrument during the campaign was monitored with a
bromobenzene internal standard. Of the two intensive observation periods (IOP) (i.e., winter:
6 January to 11 February and summer: 21 July to 23 August) data from 1 to 7 February 2012
were selected for analysis in this study. During this period the measurement site was
influenced by local sources, as well as by air masses from other parts of the UK and the
continent (Bohnenstengel et al., 2015).
A dual channel GC with flame ionisation detector (DC-GC-FID; Hopkins et al. (2003)) was
deployed at the same site as the PTR-ToF-MS during the ClearfLo IOPs. A wide range of
VOC including alkanes, alkenes, dienes, aromatic compounds and OVOC was measured (see
Table 1). The sampling time was 10 min while the analysis runtime was around 50 min,
resulting in approximately one measurement per hour.





1    Table 1: Mixing ratios, rate coefficient and OH reactivity of the VOC measured with DC-GC-

2    FID during ClearfLo from 1 – 7 February 2012.

| Compound | VMR (ppbV) | Concentration (molecules cm$^{-3}$) | $k_{OH}$ (cm$^3$ molecules$^{-1}$ s$^{-1}$) | OH reactivity (s$^{-1}$) |
|---|---|---|---|---|
| **Alkanes** | | | | |
| Ethane[a] | 12.91 ± 10.89 | (3.14 ± 2.65) x 10$^{11}$ | 2.40 x 10$^{-13}$ | 0.075 |
| Propane[a] | 4.59 ± 3.35 | (1.12 ± 0.81) x 10$^{11}$ | 1.10 x 10$^{-12}$ | 0.123 |
| iso-Butane[b] | 1.42 ± 1.00 | (3.45 ± 2.43) x 10$^{10}$ | 2.12 x 10$^{-12}$ | 0.073 |
| n-Butane[b] | 2.35 ± 1.60 | (5.71 ± 3.89) x 10$^{10}$ | 2.36 x 10$^{-12}$ | 0.135 |
| Cyclopentane[b] | 0.10 ± 0.11 | (2.50 ± 2.58) x 10$^9$ | 4.97 x 10$^{-12}$ | 0.012 |
| iso-Pentane[b] | 0.83 ± 0.62 | (2.03 ± 1.50) x 10$^{10}$ | 3.60 x 10$^{-12}$ | 0.073 |
| n-Pentane[b] | 0.42 ± 0.26 | (1.02 ± 0.64) x 10$^{10}$ | 3.80 x 10$^{-12}$ | 0.039 |
| 2,3-Methylpentane[b*] | 0.35 ± 0.29 | (8.56 ± 6.93) x 10$^9$ | 3.10 x 10$^{-11}$ | 0.265 |
| n-Hexane[b] | 0.13 ± 0.09 | (3.16 ± 2.29) x 10$^9$ | 5.20 x 10$^{-12}$ | 0.016 |
| n-Heptane[b] | 0.09 ± 0.07 | (2.18 ± 1.58) x 10$^9$ | 6.76 x 10$^{-12}$ | 0.015 |
| 2,2,4 TMP[b] | 0.04 ± 0.02 | (9.88 ± 5.22) x 10$^8$ | 3.34 x 10$^{-12}$ | 0.003 |
| n-Octane[b] | 0.03 ± 0.02 | (6.75 ± 3.75) x 10$^8$ | 8.11 x 10$^{-12}$ | 0.005 |
| | | | | |
| **Alkenes** | | | | |
| Ethene[a] | 1.93 ± 1.04 | (4.68 ± 2.52) x 10$^{10}$ | 7.80 x 10$^{-12}$ | 0.365 |
| Propene[a] | 0.43 ± 0.30 | (1.05 ± 0.73) x 10$^{10}$ | 2.90 x 10$^{-11}$ | 0.306 |
| trans-2-Butene[b] | 0.04 ± 0.03 | (1.03 ± 0.81) x 10$^9$ | 6.40 x 10$^{-11}$ | 0.066 |
| 1-Butene[b] | 0.08 ± 0.05 | (1.90 ± 1.21) x 10$^9$ | 3.14 x 10$^{-11}$ | 0.060 |
| iso-Butene[a] | 0.11 ± 0.07 | (2.63 ± 1.77) x 10$^9$ | 5.10 x 10$^{-11}$ | 0.134 |
| cis-2-Butene[b] | 0.03 ± 0.02 | (6.92 ± 5.72) x 10$^8$ | 5.64 x 10$^{-11}$ | 0.039 |
| trans-2-Pentene[b] | 0.04 ± 0.03 | (9.13 ± 7.37) x 10$^8$ | 6.70 x 10$^{-11}$ | 0.061 |
| 1-Pentene[b] | 0.03 ± 0.02 | (7.32 ± 5.27) x 10$^8$ | 3.14 x 10$^{-11}$ | 0.023 |
| | | | | |
| Acetylene[a] | 1.43 ± 0.74 | (3.47 ± 1.81) x 10$^{10}$ | 7.50 x 10$^{-13}$ | 0.026 |
| | | | | |
| **Dienes** | | | | |
| Propadiene[b] | 0.02 ± 0.01 | (4.40 ± 2.61) x 10$^8$ | 9.82 x 10$^{-12}$ | 0.004 |
| 1,3-Butadiene[b] | 0.05 ± 0.03 | (1.14 ± 0.76) x 10$^9$ | 6.66 x 10$^{-11}$ | 0.076 |
| Isoprene[a] | 0.02 ± 0.02 | (5.37 ± 4.07) x 10$^8$ | 1.00 x 10$^{-10}$ | 0.054 |
| | | | | |
| **Aromatic compounds** | | | | |
| Benzene[a] | 0.41 ± 0.17 | (9.88 ± 4.06) x 10$^9$ | 1.20 x 10$^{-12}$ | 0.012 |
| Toluene[a] | 0.64 ± 0.48 | (1.56 ± 1.17) x 10$^{10}$ | 5.60 x 10$^{-12}$ | 0.087 |
| Ethylbenzene[b] | 0.14 ± 0.11 | (3.48 ± 2.57) x 10$^9$ | 7.00 x 10$^{-12}$ | 0.024 |
| m+p Xylene[b*] | 0.18 ± 0.14 | (4.28 ± 3.52) x 10$^9$ | 1.87 x 10$^{-11}$ | 0.080 |
| o-Xylene[b] | 0.17 ± 0.12 | (4.02 ± 2.82) x 10$^9$ | 1.36 x 10$^{-11}$ | 0.055 |
| | | | | |
| **Oxygenated VOC** | | | | |
| Acetaldehyde[a] | 2.37 ± 1.38 | (5.77 ± 3.35) x 10$^{10}$ | 1.50 x 10$^{-11}$ | 0.866 |
| MACR[b] | 0.16 ± 0.12 | (3.89 ± 2.97) x 10$^9$ | 2.90 x 10$^{-11}$ | 0.113 |
| Methanol[a] | 1.44 ± 0.81 | (3.50 ± 1.96) x 10$^{10}$ | 9.00 x 10$^{-13}$ | 0.031 |
| Acetone[a] | 1.11 ± 0.51 | (2.69 ± 1.24) x 10$^{10}$ | 1.80 x 10$^{-13}$ | 0.005 |
| MVK[b] | 0.28 ± 0.15 | (6.72 ± 3.61) x 10$^9$ | 2.00 x 10$^{-11}$ | 0.134 |
| Ethanol[a] | 5.48 ± 3.81 | (1.33 ± 0.93) x 10$^{11}$ | 3.20 x 10$^{-12}$ | 0.426 |
| Propanol[a] | 0.31 ± 0.21 | (7.41 ± 5.15) x 10$^9$ | 5.80 x 10$^{-12}$ | 0.043 |
| Butanol[a] | 0.59 ± 0.33 | (1.45 ± 0.80) x 10$^{10}$ | 8.50 x 10$^{-12}$ | 0.123 |

a) IUPAC preferred value; b) Atkinson and Arey (2003); * Average of both



**PARADE.** For comparison, data collected with a PTR-ToF-MS (Ionicon Analytik GmbH,
Austria) (described in Jordan et al. (2009)) during the PARADE field campaign, were
analysed. Measurements were taken between 15 August and 9 September 2011 at the Taunus
observatory on the summit of Kleiner Feldberg (50°13´25" N, 8°26´56" E) under various
meteorological conditions. A detailed description of the measurement site and measurements
performed during PARADE can be found in Crowley et al. (2010) and Bonn et al. (2014).
The PTR-ToF-MS was operated continuously with minor interruptions. Background
measurements were conducted regularly with zero air and calibration measurements were
performed with a multicomponent gas standard before and after the campaign. For this study
two weeks of data (21 to 27 August 2011 - Period 1; 01 to 06 September 2011 - Period 2)
were selected, each with approximately the same amount of data points as the ClearfLo
dataset. Period 1 was mainly influenced by continental air masses and only towards the end
by air that travelled over the UK and the English Channel (UK-marine). Period 2 was
dominatet by UK-marine air, but was also influenced by air masses that travelled over the
Atlantic (see Phillips et al. (2012)).
**Data.** While the ClearfLo data presented here were collected at an urban background site with
mainly anthropogenic emissions, the PARADE campaign took part at a semi-rural site.
Biogenic emissions were expected from the direct vicinity, but some anthropogenic influence
was apparent from the proximity of the highly populated Rhein-Main area and Frankfurt.
Three mass channels were selected for the analysis corresponding to acetone/propanal,
toluene and ethylbenzene/xylene. In the following, the combined signal of acetone and
propanal is referred to as acetone as well as the signal of ethylbenzene and xylene is referred
to as xylene for more clarity. The compounds used for analysis represent different sources of
VOC. Toluene and xylene are counted along anthropogenic VOC, monoterpenes are of
biogenic origin and the OVOC (acetone and methanol) are whether emitted directly or
produced by photochemical oxidation in the atmosphere (Monks et al. (2009) and references
therein). They were selected, because their volume mixing ratios could be determined with
low uncertainty for both instruments. Aromatic compounds such as toluene and xylene are
well suited for this investigation, because they often show short-term high variability. The
analysis of the PARADE data also includes methanol and the sum of monoterpenes. The
characteristic parameters of the measurements during ClearfLo and PARADE are given in
Table 2.





Table 2: Characteristics of the two different PTR-ToF-MS deployed during ClearfLo and
PARADE. Given are the sensitivity based on normalised counts per second (ncps), accuracy
as error for the measurements and the limit of detection (LOD), which was calculated as $1\sigma$
for ClearfLo and $2.6\sigma$ for PARADE based on 1 min data.

|  | Compound | Sensitivity (ncps ppbV$^{-1}$) | Accuracy (%) | LOD ($1\sigma$) (ppbV) |
|---|---|---|---|---|
| ClearfLo | Acetone | 9.89 | 18* | 0.56 |
|  | Toluene | 6.36 | 18* (22) | 0.38 |
|  | Xylene | 9.00 | 18* (20) | 0.41 |

* 1st column does not include effect of isobaric overlap from aromatic fragmentation, 2nd column includes estimation of isobaric overlap.

|  | Compound | Sensitivity (ncps ppbV$^{-1}$) | Accuracy (%) | LOD ($2.6\sigma$) (ppbV) |
|---|---|---|---|---|
|  | Acetone | 37.0 | 16 | 0.08 |
|  | Toluene | 26.9 | 8 | 0.04 |
| PARADE | Xylene | 33.4 | 13 | 0.01 |
|  | Methanol | 12.7 | 17 | 0.24 |
|  | Monoterpenes | 14.1 | 10 | 0.02 |

Effects of isobaric overlap from fragmentation taken into account.

Figure 1 shows the time series of the VOC for ClearfLo (top) and PARADE Period 1
(bottom). The range of mixing ratios for ClearfLo is much wider and higher mixing ratios are
reached. Values for acetone show values up to a factor of 1.8 higher in ClearfLo compared to
PARADE, while the aromatic compounds are two orders of magnitude higher. This
emphasises the diversity of the two field sites. In the box plots, presented in Figure 2, some
interesting patterns are apparent. For ClearfLo all three compounds exhibit a similar
interquartile range (0.60 to 0.86 ppbV) but also very high maximum values. For PARADE a
different distribution is depicted. Acetone has a wider interquartile range of 1.83 ppbV and
has a higher mean value than toluene and xylene. The aromatic compounds have a much
smaller range compared to ClearfLo (0.03 to 0.08 ppbV).  Methanol has a wider range than
acetone and the monoterpenes look similar to the aromatic compounds. Both periods of
PARADE show the same pattern. The ranges of the mixing ratios during the campaigns are





summarised in Table 3. Values below the limit of detection (LOD) as well as negative values
are not disregarded in this analysis to preserve the full range of the data in order that they can
be compared to a randomly generated dataset.

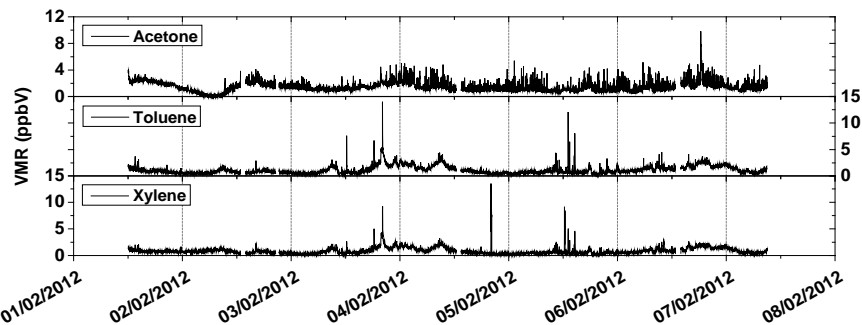

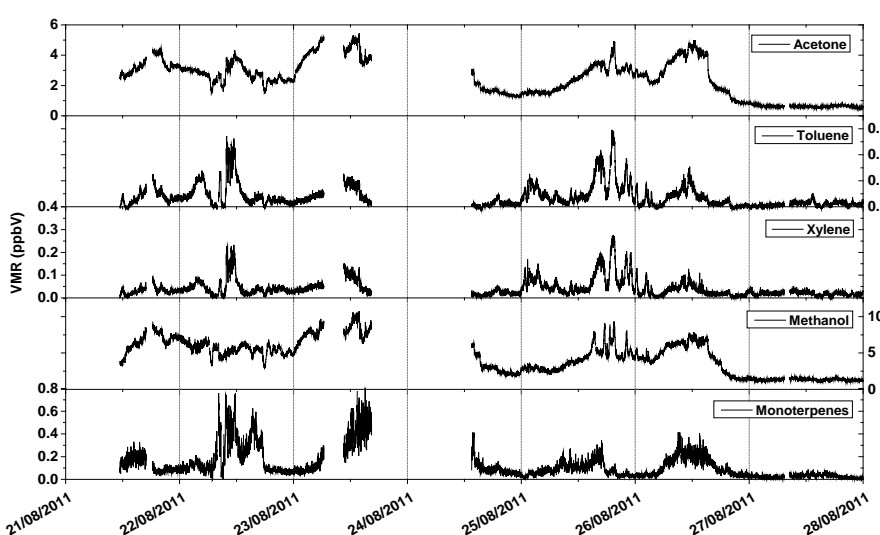

Figure 1: Time series of VOC during ClearfLo (top) and PARADE Period 1 (bottom). The
time resolution is 1 min.





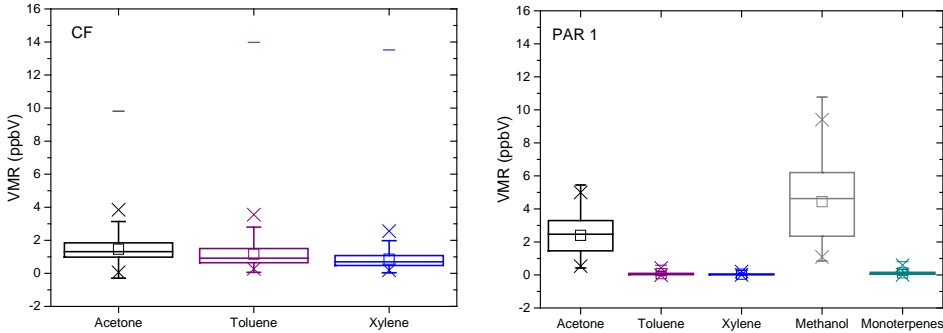

Figure 2: Box plots for ClearfLo (left) and PARADE Period 1 (right) showing the minimum,
maximum, mean (□), median, interquartile range (box) and percentiles at 1% and 99% (×).
Table 3: Overview of the range of VOC mixing ratios during ClearfLo and PARADE (PAR).

|          |           | Minimum | Maximum | Mean  | Interquart. Range | Max - Min |
|----------|-----------|---------|---------|-------|-------------------|-----------|
|          | Acetone   | -0.294  | 9.816   | 1.459 | 0.864             | 10.110    |
| ClearfLo | Toluene   | 0.058   | 13.982  | 1.162 | 0.862             | 13.924    |
|          | Xylene    | 0.038   | 13.519  | 0.861 | 0.601             | 13.482    |
|          | Acetone   | 0.426   | 5.447   | 2.400 | 1.833             | 5.021     |
|          | Toluene   | -0.030  | 0.592   | 0.076 | 0.078             | 0.622     |
| PAR 1    | Xylene    | -0.008  | 0.277   | 0.041 | 0.030             | 0.285     |
|          | Methanol  | 0.851   | 10.775  | 4.438 | 3.858             | 9.923     |
|          | Monoterp. | -0.008  | 0.801   | 0.124 | 0.116             | 0.809     |
|          | Acetone   | 0.544   | 4.873   | 1.987 | 1.797             | 4.329     |
|          | Toluene   | -0.017  | 0.646   | 0.078 | 0.073             | 0.663     |
| PAR 2    | Xylene    | -0.004  | 0.358   | 0.046 | 0.039             | 0.362     |
|          | Methanol  | 0.781   | 10.649  | 3.776 | 2.739             | 9.869     |
|          | Monoterp. | -0.009  | 0.692   | 0.075 | 0.086             | 0.701     |

OH reactivity relating to the VOC under study is calculated from the first term of equation
(1). Reaction rates for acetone ($1.8 \times 10^{-13}$ cm$^3$ molecule$^{-1}$ s$^{-1}$), toluene ($5.6 \times 10^{-12}$ cm$^3$ molecule$^{-}$





1 $^{1}$ s$^{-1}$), methanol (9.0x10$^{-13}$ cm$^{3}$ molecule$^{-1}$ s$^{-1}$) and α-pinene (5.3x10$^{-11}$ cm$^{3}$ molecule$^{-1}$ s$^{-1}$) are

2 taken from http://iupac.pole-ether.fr/index.html. The exact composition of the monoterpene

3 signal is not known, thus only the reaction rate of α-pinene is used. For xylene the average of

4 the reaction rates of ethylbenzene and o-, m- and p-xylene (14.5x10$^{-12}$ cm$^{3}$ molecule$^{-1}$ s$^{-1}$)

5 (Atkinson and Arey, 2003) was applied. Table 4 summarises the minimum, maximum and

6 mean reactivity calculated from these VOC as described.

8 Table 4: Minimum, maximum and mean VOC reacitvity and standard deviation calculated
9 from the VOC under study for ClearfLo and PARADE.

| VOC reactivity (s^-1) | | Minimum | Maximum | Mean | Stdev |
|---|---|---|---|---|---|
| CF | ATX | 0.036 | 4.864 | 0.463 | 0.289 |
| PAR1 | ATX | 0.000 | 0.191 | 0.035 | 0.028 |
| | ATX+M+MT | 0.026 | 1.296 | 0.292 | 0.205 |
| PAR2 | ATX | 0.001 | 0.222 | 0.036 | 0.024 |
| | ATX+M+MT | 0.044 | 0.987 | 0.215 | 0.119 |

ATX: Acetone, toluene and xylene

ATX+M+MT: Acetone, toluene, xylene, methanol and monoterpenes

11 The time resolution of PTR-ToF-MS is only limited by the signal to noise ratio and resulting

12 detection limit. Both instruments were operated with a 1 min time resolution. Volume mixing

13 ratios of VOC were averaged over different intervals and standard deviations were derived.

14 An average was only included for analysis, if its data recovery was at least 50%. The OH

15 reactivity for each VOC was calculated and summed as required. Only the standard deviations

16 were propagated as errors of reactivity, as the focus of this work is on investigating VOC

17 variability.

18 For clarity throughout this paper the notation R = $R_{OH}$ for reactivity regarding the OH radical

19 replaces $k_{OH}$ (cf., eq. (1); see also Nölscher et al. (2012a)). Indices denote the origin of the

20 data (PTR = PTR-ToF-MS or GC = DC-GC-FID and CL = ClearfLo or PAR = PARADE).

21 Numbers indicate the averaging time in minutes. If only some VOC are taken into account for

22 calculating the reactivity, this will be indicated, e.g. $R_{PTR,CL}^{OVOC,5}$ is the OH reactivity calculated

23 from the 5 min mean concentration of acetone, measured with the PTR-ToF-MS during

24 ClearfLo.





## 2.2 VOC reactivity distribution
For a more general view of the factors that drive variation in VOC reactivity, its frequency
distribution was investigated. GC data from the winter (9 January – 9 February 2012) and
summer IOP (18 July – 19 August 2012) during ClearfLo were applied. The VOC reactivity,
$R_{GC,CL}^{VOC_i}$, was calculated for each measured VOC$_i$ and ranged from 0.003 to 0.822 s$^{-1}$ in winter
and from 0.001 to 1.568 s$^{-1}$ in summer with a total VOC reactivity of 4.010 s$^{-1}$ and 3.862 s$^{-1}$,
respectively. The majority of $R_{GC,CL}^{VOC_i}$ values lies below 0.1 s$^{-1}$ as can be seen from the
frequency distribution plotted in Figure 3, where more than 70% of the winter and 80% of the
summer data are in the first interval from 0 to 0.1 s$^{-1}$. Seasonal differences in OH reactivity
emission rates have previously been described by Nölscher et al. (2013) for measurements at
a Norway spruce between spring and early autumn. Although the composition of VOC during
ClearfLo changed from winter to summer, no seasonal dependency could be found in the
shape of the frequency distribution for the reactivity $R_{GC,CL}^{VOC_i}$. In both cases $R_{GC,CL}^{TVOC}$ is
dominated by the sum of low reactivity contributions and less by single compounds with high
reactivity.

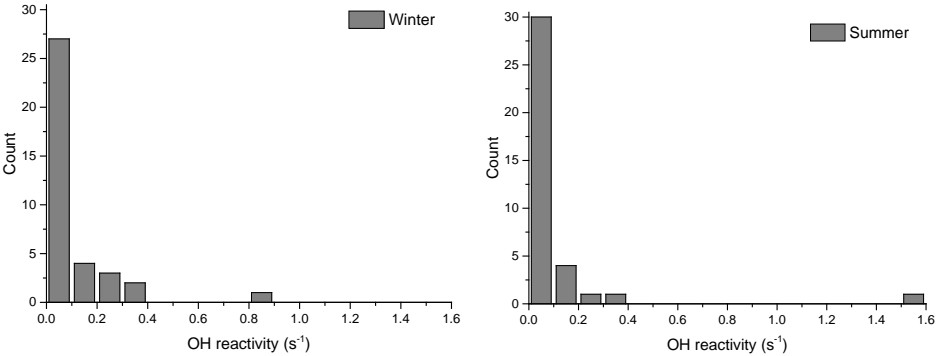

Figure 3: OH reactivity, $R_{GC,CL}^{VOC_i}$, frequency distribution for the ClearfLo campaign in winter
(left) and summer (right). Bin size is 0.1 s$^{-1}$ for both plots.
## 2.3 Generation of a randomized data set
To differentiate between pure statistical effects and measurement related characteristics, a
randomized data set was produced and analysed in the same way as the PTR-ToF-MS data.
The distribution of OH reactivity is skewed towards smaller values and only positive values



of OH reactivity are expected, hence it is better described by a log-normal distribution
compared to a normal distribution (Limpert et al., 2001). The data set of random numbers was
generated simulating an OH reactivity distribution comparable to the ClearfLo data set. The
sample mean m = 0.463 s$^{-1}$ and standard deviation sd = 0.289 s$^{-1}$ from the ClearfLo 1 min
dataset were used to define the parameters μ (equation (2)) and σ (equation (3)) for the log-
normal distribution of random numbers.

$$\mu = \log\left(\frac{m}{\sqrt{1+\dfrac{sd^2}{m^2}}}\right) \tag{2}$$

$$\sigma = \sqrt{\log\left(1+\frac{sd^2}{m^2}\right)} \tag{3}$$

A log-normal distribution of a total of 8040 random numbers was generated using the dlnorm
(#, μ, σ)-function in R. This provides a set of data comparable to 134 hours of OH reactivity
measurements with a time resolution of 1 min. Figure 4 shows the random data set as a time
series together with the hourly mean containing 60 data points. On observation of Figure 4, it
becomes obvious that the range of the hourly average is very small with a standard deviation
of 0.034 s$^{-1}$.

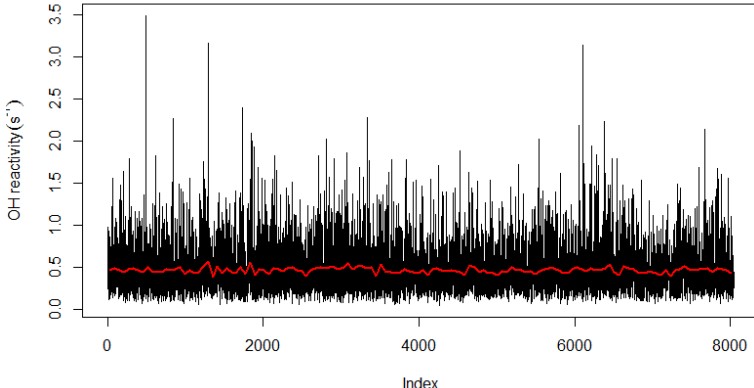

Figure 4: Time series of randomly generated log-normal data set containing 8040 numbers.
The "1min" data are shown in black. The average over 60 data points is plotted in red.





**3    Results and Discussion**
Initially the OH reactivity $R_{PTR}^{VOC}$ was calculated from the PTR data for ClearfLo and
PARADE. For both campaigns the signals of acetone, toluene and xylene (referred to
generally as VOC) were used. The effects of differing sampling intervals on the derived
reactivity were explored. For each campaign and VOC dataset a correlation of the average
values of $R_{PTR}^{VOC,t}$ for different intervals (t = 5, 10, 20 and 30 min) against the 60 min average
$R_{PTR}^{VOC,60}$ was calculated. The intervals are chosen to be the first t minutes of each hour to
simulate the initiation of a GC sequence, thus the 10 min average also covers the 5 min
averaging period and so on.
Figure 5 shows the linear correlation of the 5 min average $R_{PTR,CL}^{VOC,5}$ versus the 60 min value
$R_{PTR,CL}^{VOC,60}$ for the ClearfLo winter campaign. Data were fitted with a bivariate regression line
with an intercept (bvf) and forced through the origin (bvfo). The deviation from the slope of
the linear regression to a unity gradient $m_{res} = \left( m_{R^{<60}/R^{60}} - 1 \right)$ is taken as a measure of how well
the value of hourly OH reactivity is represented by the shorter interval average and is further
referred to as the residual slope.
The slopes of both fits in Figure 5 are below 1.0, indicating an under prediction of the
reactivity during ClearfLo by the value calculated from the first 5 min of each hour. In this
case there is only a small deviation (1.3%) from a unity gradient (see Figure 6). For all
averaging intervals the slope is equal to 1 in the range of the uncertainties of the fit.
**3.1    Effects of different sampling intervals**
For the different averaging intervals the difference to the hourly average
($\Delta R = R_{PTR,CL}^{VOC,t<60} - R_{PTR,CL}^{VOC,t=60}$) was calculated and their standard deviations are given in Table 5
as a measure of variance. $\Delta R$ generally decreases with increasing averaging time. Also
presented in Table 5 are the results from fitted Gaussian functions to the frequency
distribution of the ratio of the shorter interval averages to the 60 min average. Bins of 0.1
were chosen for the frequency distributions. The standard deviation of $\Delta R$ as well as the full
width at half maximum (FWHM) decrease, when averages are calculated for longer intervals.
The centre of all Gaussian fits achieve 0.99.





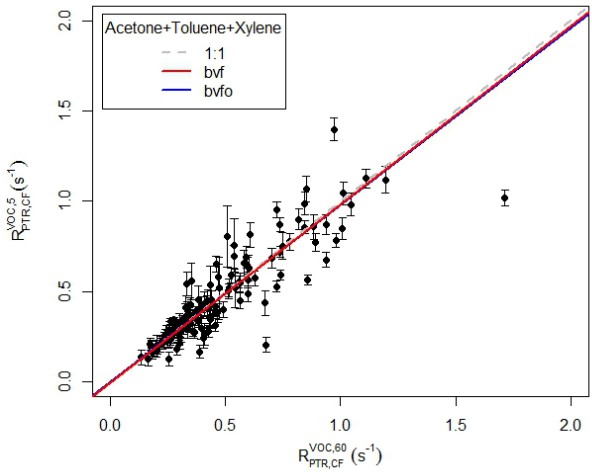

Figure 5: Linear correlation with bivariate fit of the OH reactivity calculated from the signals
of acetone, toluene and xylene   for average intervals of 5 min and 60 min for ClearfLo. The
standard deviation of the 5 min means are plotted as error bars.

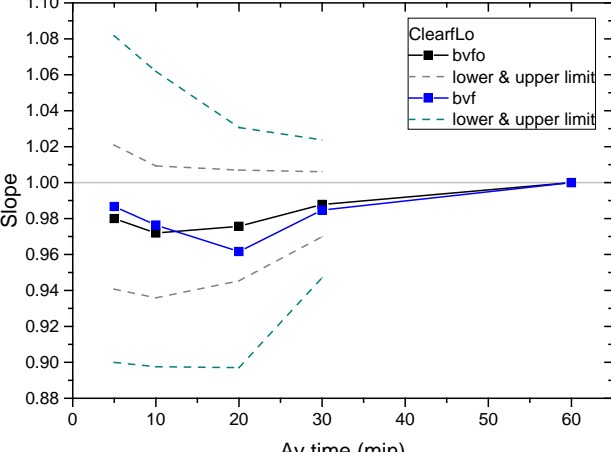

Figure 6: Development of the slope of the correlation of OH reactivity $R_{PTR,CL}^{VOC,t}$ depending on
the sampling interval for ClearfLo. Slopes for bvfo (black) and bvf (blue) with their lower and
upper limits are shown.





Table 5: Standard deviation of $\Delta R$ and results from Gaussian fits of the ratio of OH reactivty
$R_{PTR,CL}^{VOC,t<60} \big/ R_{PTR,CL}^{VOC,t=60}$ calculated from shorter interval averages to 60 min average for ClearfLo.

| Notation | Time interval (min) | $\Delta R$ Stdev $(s^{-1})$ | Gaussian Fit Centre | FWHM |
|---|---|---|---|---|
| $R_{PTR,CL}^{VOC,5}$ | 5 | 0.12 | $0.998 \pm 0.011$ | $0.337 \pm 0.025$ |
| $R_{PTR,CL}^{VOC,10}$ | 10 | 0.12 | $0.997 \pm 0.008$ | $0.244 \pm 0.020$ |
| $R_{PTR,CL}^{VOC,20}$ | 20 | 0.10 | $0.988 \pm 0.006$ | $0.246 \pm 0.013$ |
| $R_{PTR,CL}^{VOC,30}$ | 30 | 0.06 | $0.992 \pm 0.004$ | $0.198 \pm 0.009$ |

For Period 1 of the PARADE data (PAR1) the results show a slope greater than 1 (Figure 7).
The high variability of the data is reflected by a higher divergence of the slopes of 1.13 for
bvf fit and 1.05 for the bvfo fit based on 5 min averaged data. The small standard deviations
of $\Delta R$ given in Table 6 highlight the narrow range of calculated OH reactivity $R_{PTR,PAR1}^{VOC}$
However, the high variability of the data is reflected by the FWHM of the frequency
distributions of the ratios which is higher for each interval when compared to ClearfLo.

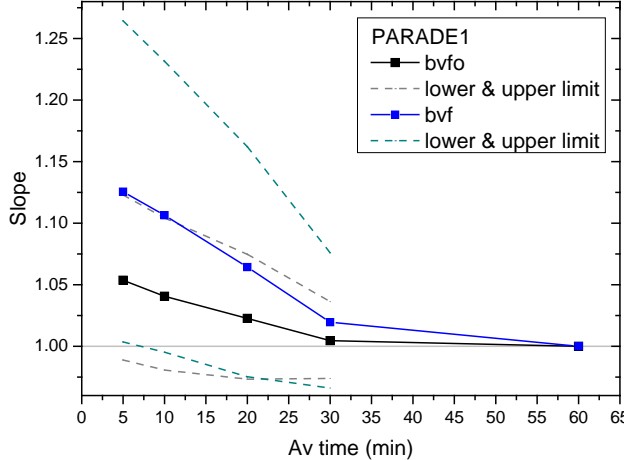

Figure 7: Development of the slope of the correlation of OH reactivity $R_{PTR,PAR1}^{VOC,t}$ depending on
the averaging time for PARADE - Period 1.





Table 6: Standard deviation of $\Delta R$ and results from Gaussian fits of the ratio of OH reactivity
$R_{PTR,PAR1}^{VOC,t<60} \big/ R_{PTR,PAR1}^{VOC,t=60}$ calculated from shorter interval averages to 60 min average for PARADE
- Period 1.

| Notation | Time interval (min) | $\Delta R$ Stdev ($s^{-1}$) | Gaussian Fit Center | FWHM |
|---|---|---|---|---|
| $R_{PTR,PAR1}^{VOC,5}$ | 5 | 0.016 | $0.980 \pm 0.011$ | $0.379 \pm 0.027$ |
| $R_{PTR,PAR1}^{VOC,10}$ | 10 | 0.015 | $0.976 \pm 0.012$ | $0.353 \pm 0.026$ |
| $R_{PTR,PAR1}^{VOC,20}$ | 20 | 0.012 | $0.997 \pm 0.013$ | $0.310 \pm 0.030$ |
| $R_{PTR,PAR1}^{VOC,30}$ | 30 | 0.008 | $0.995 \pm 0.009$ | $0.273 \pm 0.020$ |

For Period 2 (PAR2), an over prediction of the OH reactivity $R_{PTR,PAR2}^{VOC}$ can be observed again
(Figure 8), but with an even greater slope of 1.26. In both periods of PARADE the slope
approaches a value of 1 as increasing averaging time is taking more of the variability within
one hour into account. Standard deviations of $\Delta R$ and FWHM values are similar to Period 1
of the PARADE data, while the centres of the Gaussians are closer to 1 (Table 7).

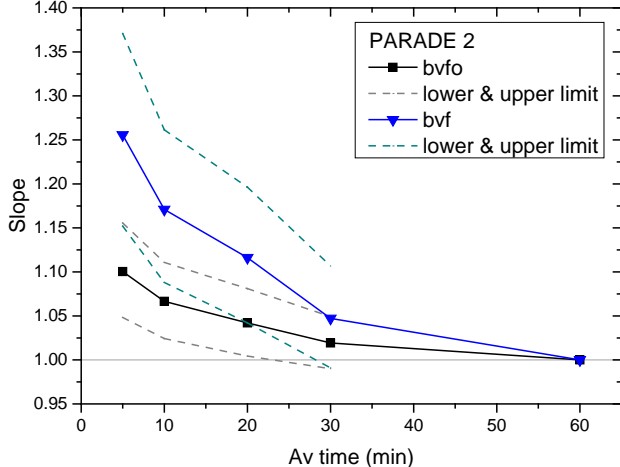

Figure 8: Development of the slope of the correlation of OH reactivity $R_{PTR,PAR2}^{VOC,t}$ depending on
the averaging time for PARADE - Period 2.





Table 7: Standard deviation of $\Delta R$ and results from Gaussian fits of the ratio of OH reactivity
$R_{PTR,PAR2}^{VOC,t<60} \big/ R_{PTR,PAR2}^{VOC,t=60}$ calculated from shorter interval averages to 60 min average for PARADE
- Period 2.

| Notation | Time interval (min) | $\Delta R$ Stdev $(s^{-1})$ | Gaussian Fit Center | FWHM |
|---|---|---|---|---|
| $R_{PTR,PAR2}^{VOC,5}$ | 5 | 0.013 | $0.996 \pm 0.014$ | $0.352 \pm 0.034$ |
| $R_{PTR,PAR2}^{VOC,10}$ | 10 | 0.009 | $0.994 \pm 0.013$ | $0.296 \pm 0.031$ |
| $R_{PTR,PAR2}^{VOC,20}$ | 20 | 0.008 | $0.992 \pm 0.008$ | $0.238 \pm 0.019$ |
| $R_{PTR,PAR2}^{VOC,30}$ | 30 | 0.006 | $1.010 \pm 0.004$ | $0.238 \pm 0.010$ |

When OH reactivity is calculated from GC measurements of VOC, some of the variability in
the data is not captured, because air sampling alternates with the GC run itself (Hopkins et al.,
2003). In this manner, the analytes are collected for a short duration which is then used to
represent the whole measurement cycle. This work suggests that a discrepancy between 60
min averages and shorter intervals can be caused due to the variable nature of atmospheric
VOC. A sampling time of only five minutes can cause a deviation of more than 25%.
Accordingly, this would then artificially contribute to missing OH reactivity.
The deviation is greater for the semi-rural measurements in the Taunus during PARADE
compared to the urban measurements in London. Although the range of the analysed VOC
reactivity is smaller during PARADE, the highly frequent fluctuations cause a greater
variability in OH reactivity for the investigated intervals.
**3.2   The distribution of residual slopes across consecutive 5 min intervals**
In the previous section, only reactivity calculated from the average of the first 5, 10, 20 and
30 min was compared to the hourly mean. Naturally, these averages have different values,
depending on the point at which they are selected from the hour under study. They may over-
or under predict the hourly mean as can be seen from Figure 9 where residual bvf slopes
between $R_{PTR}^{VOC,5}$ and $R_{PTR}^{VOC,60}$ (cf. Figure 4) are plotted for consecutive 5 min averaging periods
within the hour. A tendency towards an over prediction of OH reactivity is observed for both





campaigns (ClearfLo - top left, PARADE – bottom) and also for the randomized data set (top
right). For the randomized data set bvfo was used - bvf has a much higher slope as the data
are clustered together within a small range. On average the residuals are nearly 10% with a
standard deviation of 0.1% or less (8.6% ± 0.1% - for ClearfLo; 8.85% ± 0.03% for PARADE
1; 9.5% ± 0.1% for PARADE 2; 4% ± 4% for the randomized data).

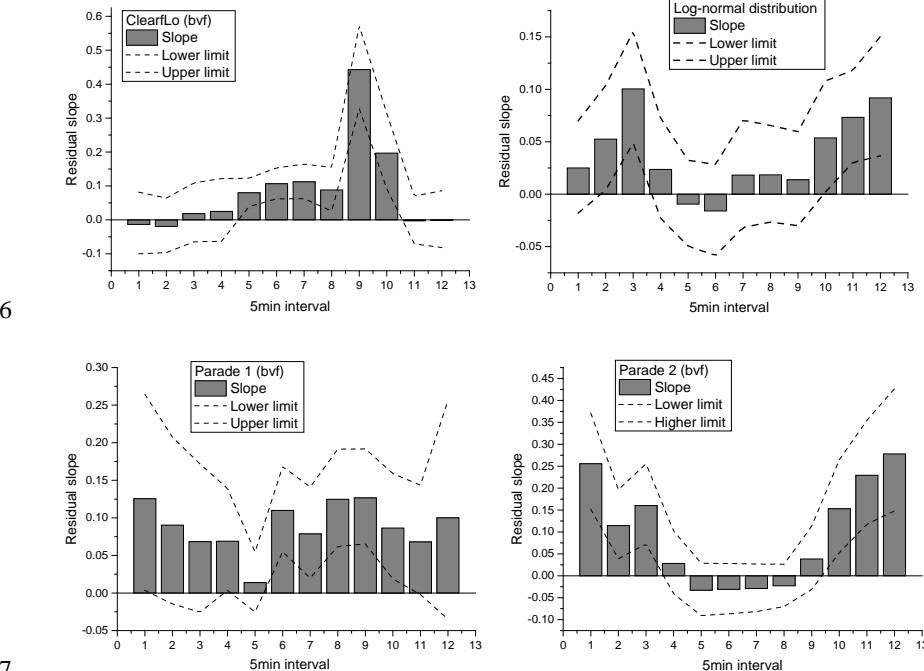

Figure 9: Residual slopes of the correlation of all 5min means to the hourly mean.
For linear regression the standard least squares fit is widely used. This method is less
adequate when errors in both y and x are assumed or when the assignment of the independent
variable is not clear (Isobe et al., 1990). Other methods for bivariate fitting in natural sciences
have been discussed in the literature (Isobe et al., 1990;Warton et al., 2006;Cantrell, 2008).
Cantrell (2008) found that a bivariate fit is less sensitive to outliers compared to an ordinary
least squares (ols) fit. Warton et al. (2006) described the major axis (ma) and standard or
reduced major axis regression (sma/rma). These methods are preferred when the agreement
between two measurement techniques is investigated. For equally important deviations from
the regression line in the x and y directions ma is used, while sma can be used when the scales



in x and y are not comparable. These two functions are implemented in the *smatr*-package in
R. The ma-function is used to produce the bivariate regression line (bvf) and the bivariate
regression forced through the origin (bvfo) in this work. In the work of Isobe et al. (1990) the
ordinary least square regression, major axis and reduced major axis regression, and
additionally ols bisector (ols-bis) regression, are compared. They point out that different
slopes are to be expected for all the bivariate fits (ma, sma, ols-bis). For ma they find large
uncertainties for the slope. To carry out a symmetrical analysis they recommend using the ols-
bisector regression.
Figure 10 shows the residual slopes between $R_{PTR,CL}^{VOC,5}$ and $R_{PTR,CL}^{VOC,60}$ for consecutive 5 min
intervals of the ClearfLo data using the different regression methods (ols, ols-bis, bvfo, bvf).
The mean of the residual ratios (i.e., the average ratio minus 1) of $R_{PTR,CL}^{VOC,5}$ to $R_{PTR,CL}^{VOC,60}$ is also
shown in Figure 10. The bvfo puts more weight onto low OH reactivity values compared to
bvf and produces a line matching the majority of the data much better. Therefore, smaller
residuals are observed compared to the bvf. The very small residual of the average ratio also
emphasize that deviation from the ideal slope of 1 is mainly driven by outliers. The ols-bis
regression shows a negative residual for all 5 min intervals. Mean deviations and ranges for
all regression methods based on consecutive 5 min averaging periods are summarised in
Table 8, where it can clearly be seen that once averaged across 12 intervals ols and the ratio
have a negligible deviation. On average the ols shows the smallest deviation from the ideal
slope of one, but in terms of stability across all 5 min intervals the ols-bis performs better.
This analysis shows, that the extend of under or over predicting OH reactivity by short
sampling intervals is a matter of how the data are compared to each other.





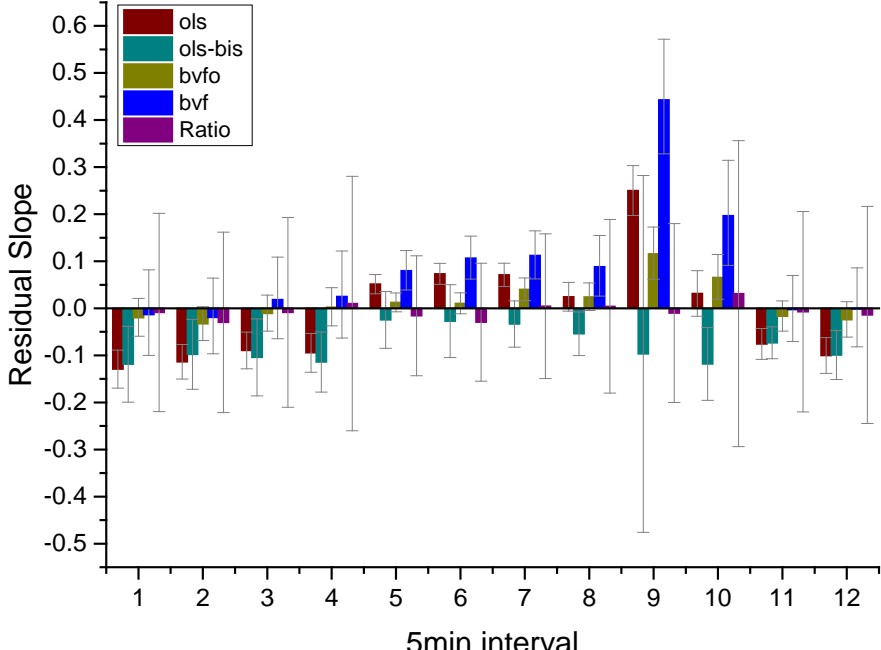

Figure 10: Residual slopes from different linear regression methods and the mean residual

ratio for all 5min intervals. Error bars depict the standard error of the slope for the ols fit, the

square root of the variance for the ols-bis and the lower and upper limit for the bvfo and bvf

fits and the standard deviation of the ratios.

Table 8: Summary of the statistics of the residual slopes and ratios from the comparisons of

the 5min means to their 60min means for the ClearfLo data.

| Method | Min | Max | Range | Mean | Stdev |
|---|---|---|---|---|---|
| ols | -0.129 | 0.250 | 0.379 | -0.008 | 0.112 |
| ols-bis | -0.119 | -0.024 | 0.094 | -0.080 | 0.036 |
| bvfo | -0.033 | 0.116 | 0.149 | 0.014 | 0.043 |
| bvf | -0.019 | 0.443 | 0.462 | 0.861 | 0.130 |
| Ratio | -0.030 | 0.031 | 0.061 | -0.006 | 0.017 |





The same analysis was performed with an extended data set that included ten times the
number of data points of a randomized log-normal distribution to test for any artefacts relating
to the limited sample size of the PTR-ToF-MS data. No appreciable difference was obtained
when compared to the smaller data set. Hence, we conclude that the observed bias to an
overestimation for the bivariate fits and an underestimation for the ols-bisector regression on
average is real and not an artefact caused by computing a shorter time series.

## 3.3  At what sampling interval can the hourly mean be represented with a smaller sub sample?

The question being further investigated here is: how many data points are needed to calculate
an average value that represents the hourly mean within its standard deviation? The ClearfLo
dataset of OH reactivity, based on acetone, toluene and xylene, was used to calculate 60 min
means of consecutive 1 min data. Small gaps in the time series were skipped such that 60
contiguous data points were computed. However, data was discarded if it included larger
gaps, e.g., 1 hour or more. The set of 60 data points was further subdivided into smaller
intervals to calculate means of OH reactivity $R_{PTR,CL}^{VOC,t<60}$ of 2, 3, 5, 10, 15, 20 and 30 min.
Residual reactivities for these averages were calculated by subtracting the hourly mean
$R_{PTR,CL}^{VOC,60}$ before plotted against the number of data points n, which in this case corresponds to
minutes (Figure 11). Corresponding standard deviations were calculated for each 60 min
mean, but only the minimum and maximum values are plotted as dashed and solid grey lines
in Figure 11, respectively. Additionally, two models are plotted, describing the course of the
functions $f_1$ (1/n) (light blue) and $f_2$ ($1/\sqrt{n}$) (dark blue) starting at the maximum and
minimum value (both marked as red dots). The positive range of residual OH reactivity is
much wider than the negative range and is capped by the $1/\sqrt{n}$-function. The negative values
show a slower approach to the mean. The 20 min averages all lie within the maximum
standard deviation, but even when averaging over 30 min the range is much wider than the
minimum standard deviation of OH reactivity.





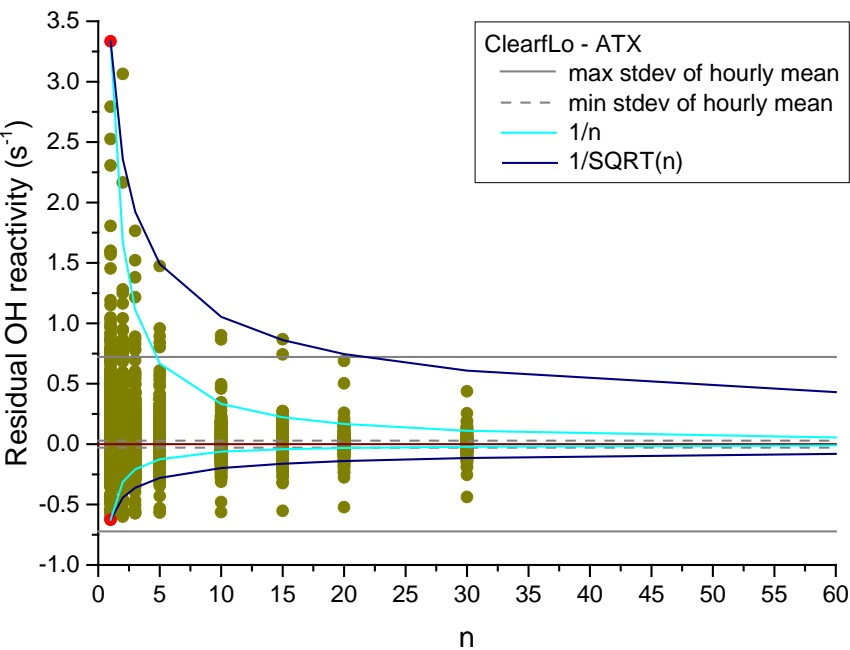

Figure 11: Dependency of the deviation in OH reactivity from the hourly mean   on the
number of data points for the entire ClearfLo data set.
The 2, 3, 5, 10, 20 and 30min averages are now compared directly to their hourly mean and
standard deviation to summarises the findings from Figure 11. As can be seen in Table 9 at
20min still 2.78% of the ClearfLo data exceed their hourly mean. At 30min all data lie within
the range of the standard deviation. Therefore, a sampling time greater than 20 min would be
required to represent the hourly mean. The random data reach a comparable level of data
exceeding the hourly mean by 2.80% for averaging over 5min only. Here, sampling for only
10 min would be sufficient for representing an hour worth of data.





Table 9: Comparison to the hourly averages and their standard deviation for ClearfLo and a
data set of log-normal distributed random numbers. Listed are the number and the percentage
of data that exceed the stdev of the hourly mean for different n. n refers to the number of
minutes that were averaged in each case.

|  | ClearfLo | | | Random numbers | | |
|---|---|---|---|---|---|---|
| n | # data | # data > stdev | % of data > stdev | # data | # data > stdev | % of data > stdev |
| 2 | 3960 | 838 | 21.16 | 4020 | 534 | 13.28 |
| 3 | 2640 | 457 | 17.31 | 2680 | 199 | 7.43 |
| 5 | 1584 | 225 | 14.20 | 1608 | 45 | 2.80 |
| 10 | 792 | 80 | 10.10 | 804 | 0 | 0 |
| 15 | 528 | 38 | 7.20 | 536 | 0 | 0 |
| 20 | 396 | 11 | 2.78 | 402 | 0 | 0 |
| 30 | 264 | 0 | 0 | 268 | 0 | 0 |

**3.4  Effect of different VOC classes on OH reactivity**
Many different atmospheric VOC have been identified (Goldstein and Galbally, 2007), all of
which contribute to OH reactivity. Based on their chemical characteristics they are often
divided into different classes. In order to identify how the variation of individual components
contributes to the observed deviation of $R_{PTR}^{VOC,5}$ from $R_{PTR}^{VOC,60}$ correlations between 5 min and
hourly mean reactivities were analysed for different VOC classes separately. The results are
shown in Figure 12 for ClearfLo (blue area) and PARADE (grey and green areas), where
OVOC contains the data from acetone for ClearfLo ( $R_{PTR,CL}^{OVOC,5}$ ) and acetone and methanol for
PARADE ( $R_{PTR,PAR}^{OVOC,5}$ ). The aromatics are calculated from toluene and xylene and BVOC refers
to the sum of the monoterpenes, which were only available for PARADE. Again a greater
deviation from 1 is observed for the PARADE data. The OVOC show no significant deviation
from 1 for both campaigns and while the aromatics are close to 1 for ClearfLo they show a
significantly different value for PARADE with a deviation of up to 31%. Finally, BVOC
deviate from a perfect correlation by 21% for the second period of PARADE.





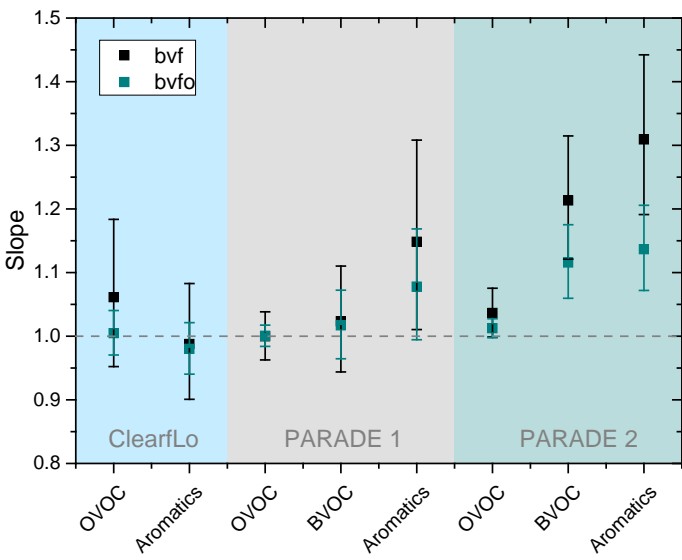

Figure 12: Bivariate fit results between 5 min averaged to 60 min averaged reactivity. Slopes
are plotted for ClearfLo (blue shaded area, left) and PARADE (Period 1 – grey shaded, Period
2 – green shaded, right). Correlations were analysed separately for OVOC (acetone for
ClearfLo and acetone and methanol for PARADE), BVOC (monoterpenes) and aromatic
compounds (toluene and xylene).
**3.5   Scaling the effect to the share of VOC reactivity during ClearfLo**
The observed deviations of the slopes from the ideal slope of 1 in Figure 12 were scaled by
their share to determine the overall effect on total VOC reactivity $R_{CL}^{TVOC}$. Data from the same
week as the PTR-ToF-MS data were used to calculate the influence of VOC speciation on OH
reactivity. Over the period of 1 to 7 February 2012 the total OH reactivity of these compounds
is $R_{GC,CL}^{TVOC} = 4.05$ s$^{-1}$. Based on Table 1, OVOC contribute most to reactivity at 43%, followed
by alkenes at 26% and alkanes at 21% of $R_{GC,CL}^{TVOC}$. The aromatic compounds have a share of
6% and dienes, including isoprene, account for 3%. Finally, the contribution of the only
measured alkyne is less than 1%.
The extend of which different VOC classes' variability effects $R_{GC,CL}^{TVOC}$ was calculated by
weighting the deviation derived from the correlations for the different classes (i.e., the
deviation of the slope between $R_{PTR,CL}^{class,5}$ and $R_{PTR,CL}^{class,60}$ from 1) by the proportion that each class





contributes to the total reactivity (calculated from Table 1). Here, it is assumed that deviations
derived from measurements of only a few compounds is representative of each class of VOC
under study.

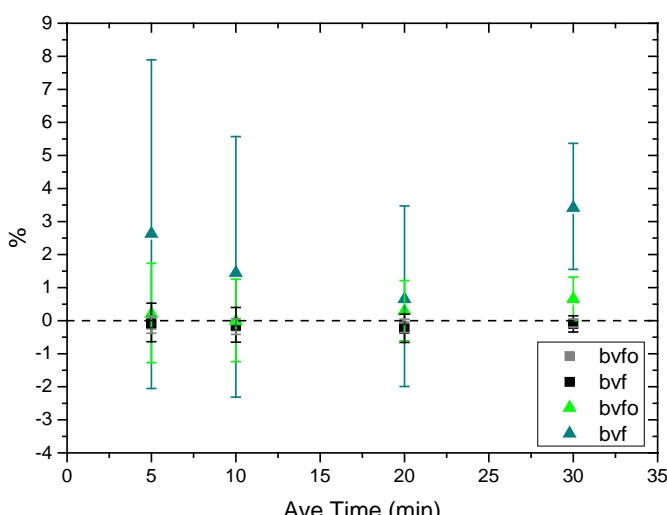

Figure 13: Percentage deviation in OH reactivity for different sampling intervals owing to
VOC variability for the ClearfLo data. Results are plotted separately for aromatic compounds
(black and grey squares) and OVOC (green triangles) for both bivariate fits without (bvfo)
and with an intercept (bvf). The deviations are based on the share of the VOC's class to total
OH reactivity for each investigated averaging interval.
Based on Figure 13 5 min averages over predict VOC reactivity by up to 2.6% due to
variability in OVOC concentrations. This value decreases for increasing averaging time, but
shows a maximum of 3.4% for the 30 min mean. There is no significant contribution of the
aromatic compounds to a deviation from the hourly mean OH reactivity for any averaging
interval.
A similar behaviour could be expected for other classes of VOC such as the alkenes and
alkanes, the second and third most important classes in Table 1. However, this could not be
tested in the present study using PTR-MS data. Yet, this study shows how the effect of using
short sampling intervals could account for a missing or overpredicted VOC reactivity in the
range of 10% or more.



Lidster et al. (2014) investigated the potential increase in OH reactivity owing to higher
substituted aromatic compounds, which are normally not measured in field campaigns. They
state that they can contribute to up to 0.9 s$^{-1}$ in VOC reactivity. This would increase the share
of aromatic compounds by more than a factor of 3, however based on the results in
Figure 13 the effect on OH reactivity would still be in the range of less than 1% while the
contribution of the OVOC would only be altered slightly.



**4    Conclusions**
The effect of using short sampling intervals for VOC measurements on resulting OH
reactivity was investigated using two different monitoring campaigns as case studies. OH
reactivity was found to be both under and over predicted due to missing variability in VOC
data. The divergence between OH reactivity calculated from 5 min sampling intervals and
hourly values was found to be around 1 - 28% and 0 - 44% for the PARADE and CleafLo
campaigns, respectively, owing to the variability of the VOC concentrations. These
discrepancies may contribute to missing OH reactivity when compared to direct
measurements. Results from the urban and the semi-rural site show on average similar effects
when comparing reactivity averaged over 5 min intervals to the hourly mean.
Comparison to a randomnized data set with a similar distribution as the CleafLo data showed
that the variability of the VOC concentations with time is the main reason for deviant results
from shorter sampling intervals. For the randomnized data a sampling time of less then 10
min is sufficient so that all data points are within the range of the hourly standard deviation,
while for the ClearfLo data it takes more than 20 min.
The effect of short sampling times of VOC concentrations on calculated OH reactivity is
differently pronounced for each VOC class. When comparing OH reactivity calculated from
VOC sampled over a 5 min period to the hourly mean, a larger divergence was found for the
aromatic compounds than OVOC during ClearfLo. The same trend was observed for the
PARADE campaign, while the effect of OVOC is almost negligible. Biogenic VOC, with the
monoterpenes as representatives, were added for analysis. They show a similar behaviour as
the OVOC, but with a slightly greater divergence.
The bigger proportion of measured OVOC, compared to the aromatic compounds, at the
urban site during ClearfLo contributes to a higher deviation in calculated OH reactivity when
using short sampling intervals. Taking the results from Lidster et al. (2014) into account, the
effect of aromatic VOC increases and but is still small.





## Acknowledgements

The authors would to thank Lis Whalley from the University of Leeds for useful comments on the manuscript. We acknowledge funding from the Natural Environment Research Council through NE/H003207/1 for ClearFlo and EU PEGASOS.



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
