# Peer review of "What effect does VOC sampling time have on derived OH"

_Atmospheric Chemistry and Physics, 2015_

## Referee Comment (RC1) · Anonymous Referee #2 · 15 Feb 2016

The manuscript "What effect does VOC sampling time have on derived OH reactivity?" presented by Sonderfeld et al. is an extended analysis of two different field campaign's high resolution data-sets with regard to the question if and how strong different sampling times and intervals impact the calculation of missing OH reactivity. This question is of relevance for understanding atmospheric photooxidation processes especially when the directly measured total OH reactivity and the budget of individually detected OH sinks such as volatile organic compounds (VOC) are compared. For example, gas chromatographic samples are often collected over a time spanning several minutes whereas the actual analysis of the sample can be as long as one hour. Sonderfeld et al. explore methodically whether a comparison between such low and high resolution data-sets might miss-represent the true variability and average value of the atmospheric observations with respect of the overall OH reactivity.
[Figure]

Overall, the data-sets discussed in this manuscript are taken from recent field measurements campaigns, the analysis has been explained carefully, the results are well presented, and the overall conclusions aim to answer the question which was raised within the title. I recommend this paper for publication in ACP and have the following minor specific, general and technical comments.

Specific comments:

SC 1) p.3, l.10- p.4, l.6: The authors present a very good compilation of studies about missing OH reactivity based on the comparison of individually measured OH sinks and the directly detected total OH reactivity. As stated later in the introduction (paragraph p.4, l.14-22) the different time resolutions of OH reactivity and individual compound measurements might bias the comparison hence the resulting missing OH reactivity. With this in mind and following the scope of the entire manuscript, it would be interesting for the reader to get some information about typical instrumentation (e.g. GC) and sampling times (e.g. first 10 min of a 40 min cycle) that have been used for VOC measurements and compared to the directly detected total OH reactivity. Is it possible to include this additional information, either for all or some of the examples that are provided in the text already?

SC 2) p.3, l.28-29: Indeed both of the studies presented here concluded that the missing OH reactivity is possibly due to unmeasured oxidation products. However, they did not exclude the contribution of undetected forest emissions that may add to the missing reactivity, as well.

SC 3) p.5, l.14-15 and p.7, l. 9: For both measurement campaigns that are chosen for presenting data in this manuscript a calibration was performed before (and for PARADE also after) the campaign. Was the calibration done at the measurement site or in the laboratory?

SC 4) Table 2: How did you calculate the accuracy as error for the measurements? Why did you chose 1sigma for the ClearfLo campaign and 2.6sigma for the PARADE
campaign as LOD?

SC 5) p.11, l.2-3: Since the different monoterpenes have very different reactivities towards OH, it is critical to know their atmospheric composition for calculating the OH reactivity due to monoterpenes. Especially the two contrasting sites presented here might have very different typical ambient monoterpene distributions resulting in different characteristic OH reactivities. Unfortunately, the PTR-MS cannot separate the different monoterpenes and detects all of them as one single signal. Therefore, the authors decided to use the reaction rate of a-pinene only. In case that during the campaigns the monoterpene composition was characterized by accompanying instrumentation (e.g. GC-MS) could you use that data to estimate a typical monoterpene OH reaction rate coefficient? Or in case that you did not have such observations during the campaign, are there any studies in the literature that could give you hints about the typical monoterpene distribution at the campaign sites? A typical monoterpene composition will help you to estimate a typical monoterpene reaction rate with OH that is representative for location and timing of the two campaigns.

SC 6) p.12, l.19: Why do you need to generate a randomized data set? What do we learn from comparing this fictive distribution of OH reactivities to the field data?

SC 7) p.14, l.20-28 and p.16 Table 5: Could you please explain why you chose to not include $\Delta R$ in Table 5? The standard deviation of $\Delta R$ is used as a measure of variance and presented in Table 5. However, I wonder if it would be more accurate to look at the standard deviation relative to the hourly average $\Delta R$? Also, could you please clarify the physical meaning of the values presented by the Gaussian Fit Centre and the FWHM (full width at half maximum)? You say that the Gaussian function was fitted on the frequency distribution of the ratio of shorter interval averages ($R(t<60)$) to the hourly average ($R(t=60min)$). In this case a value of 1 would be calculated for perfect overlap of those two averages and the FWHM would convert to zero, right?

SC 8) p. 16, l. 7-8: These two sentences seem to have contradicting statements: The

small standard deviation of $\triangle$R highlights the narrow range of calculated OH reactivity. And the high variability of the data is reflected by the relatively high FWHM. Could you please explain? Would you get a different result if looking at the relative standard deviation?

SC 9) p.18, l.12-15: A more general remark in this context: It could be interesting, regarding the discussion about the difference of variabilities at the two different sites, to have a look at the variability-lifetime relationship as for example presented in Williams et al. 2000 (http://dx.doi.org/10.1029/2000JD900203). Similarly for section 3.4 in which the effect of different VOC classes on OH reactivity is discussed.

SC 10) p. 26, Figure 13: The effect of VOC variability on the calculated OH reactivity is presented in this figure for the ClearfLo campaign. How does it look like for the PARADE campaign data? Are there significant differences?

SC 11) p. 28, l. 4-5: The missing variability in VOC data, that you mention here, is only due to the short interval sampling time. Is this correct?

SC 12) p. 28, l.5-6: The divergence between 5 min and 60 min averaged calculated OH reactivity is given here to be between 1-28% (PARADE) and 0-44% (ClearfLo). These numbers appear in the text for the first time at this point within the conclusions. Could you please include some reference in the text beforehand? And also it would be good to stronger point out the conditions and the statistical test (e.g. first 5 min of hour, consecutive 5 min intervals, regression methods, number of data points, effect of different VOC classes, . . .) that lead to the greatest divergence.

General comments:

GC 1) Within the presented study you solely compare VOC data with the OH reactivity calculation based on the measurement of individual OH sink compounds. Do you have any directly measured total OH reactivity data available to compare to?

GC 2) For the statistical analysis of the two field campaigns (ClearfLo and PARADE)

[Figure]

the entire data-set was used. How does your overall conclusion depend on the time of the day? Did you test the small sampling interval averages against the hourly averaging for example for day and nighttime data only? Is it possible that some VOC show decreased variability within the nocturnal boundary layer whereas during daytime the close distance to emission sources and turbulent mixing increase their overall variability?

Technical comments:

TC 1) p.2, l.10: "Its actual concentration being determined by the balance between its sources and sinks." It seems to me that the verb in this sentence needs to be "is" rather than "being".

TC 2) p.2, l.17: Here, a list of references about in-situ measurements of OH reactivity is provided. However, it should be indicated (e.g. with "e.g.") that this list only presents a fraction of the actual literature.

TC 3) p.3, l. 10: ". . . good agreement between measured and calculated OH reactivity have been found." It should be "has" instead of "have".

TC 4) Table 1: Are here averages or median values presented for the mixing ratios and concentrations? What is the given uncertainty? Standard deviation?

TC 5) p.7, l.25: I think you do not need the "whether" in this sentence.

TC 6) p.8, l.8: Here you repeat yourself by having "values" twice in one sentence.

TC 7) p.10 Figure 2: It would be great, if you could add a legend to the two graphs explaining the different markers used.

TC 8) p.10 Table 3: Table 3 basically repeats what is shown in Figure 2. I wonder if it is really necessary to include the same information twice. You might want to decide whether to present the figure or the table. Also, what units did you use to present the range of VOC mixing rations in Table 3?

[Figure]

TC 9) p.11 Table 4: Please correct the format of the VOC reactivity unit. Also, to be precise it is the OH reactivity due to the selected VOC.

TC 10) p.12, section heading and terminology: The term "VOC reactivity" can be misleading because atmospheric VOC typically react with various oxidants such as O3, NO3, Cl or OH. Hence, it is more precise to use the terminology OH reactivity. This applies already for most of the presented manuscript (e.g. Title, Figure 3 ect.) but should be checked for consistency, especially in this section 2.2.

TC 11) p.12, l.13: In the previous section you define the notation for different OH reactivity calculations which depends on the instrumentation, campaign and compounds taken into account. The example shows that OVOC during the ClearfLo campaign only includes acetone. Probably during the Parade campaign it would also include methanol. Then in section 2.2 the OH reactivity was calculated for VOC detected by GC during the ClearfLo campaign. However, what do you mean with TVOC as referred to in line 13?

TC 12) p.18, l. 20-21: Here, it is referred to Figure 4, which shows the randomly generated data set. In the context of presenting residual slopes as in Figure 9, I found this confusing and it might be a mistake. Also, from this point on you look at the residual slopes (as plotted in Figure 9). In the figures and sections before (e.g. Fig. 6, 7, 8) the slope was shown. Is there a reason for not being consistent about that?

TC 13) p. 24, Table 9: With "Random numbers" do you mean the "Randomly generated data set"?

---

## Referee Comment (RC2) · Anonymous Referee #1 · 15 Feb 2016

This manuscript reports how OH reactivity values calculated from trace gas measurements could be impacted by unmeasured VOC variability when VOC instruments are not capable of continuous measurements, e.g. Gas Chromatographic instruments. The authors used 1-min continuous measurements of VOCs ( PTR-ToFMS) from two different field campaigns to calculate hourly means of OH reactivity. Additional calculations of hourly values were performed by averaging VOCs over different time intervals (5 to 30 min) within each hour to mimic what would have been calculated from GC measurements characterized by various sampling durations. Overall, this study demonstrates that deviations up to 25% could be observed for targeted VOCs. This significant source of errors has to be accounted for when calculated OH reactivity values are compared to direct measurements.

This manuscript is well structured, clear and concise, and will be of interest for the

atmospheric community. I therefore recommend publication in ACP after the authors address the following minor comments:

Minor comments:

P2 L12: "...it sinks are manifold..." should read "... OH sinks are manifold..."

P5 L9: Was the Kore PTR-ToFMS equipped with an ion funnel? It seems so since the study of Barber et al. (2012) is cited. It should be clearly stated in the text and the authors should indicate, if relevant, how the ion funnel impacts the PTRMS response (sensitivity, humidity effects...).

P5 L13-14: "For background measurements a hydrocarbon trap was employed". Please provide details about this hydrocarbon trap. How efficient was it to scrub hydrocarbons. How were background measurements performed for OVOCs? How often were the background measurements recorded?

P5 L17-18: "The stability of the instrument during the campaign was monitored with a bromobenzene internal standard". Could the authors indicate how stable it was during the campaigns? Was there a need to correct for a drift in sensitivity? If so, how was it done?

P5 L25-26: Was an ozone scrubber used for the GC measurements?

P18 L10-11: "A sampling time of only five minutes can cause a deviation of more than 25%. Accordingly, this would then artificially contribute to missing OH reactivity." This reviewer does not agree with the last sentence, which should be rephrased. The deviation will either lead to a positive or negative bias and will not always appear as missing OH reactivity. This deviation should be discussed as an additional source of errors to account for when measured and calculated OH reactivity values are compared.

P18 L21: "...(cf. Figure 4)..." Wrong figure.

P23 L5-10: "As can be seen in Table 9 at 20min still 2.78% of the ClearfLo data exceed

their hourly mean. At 30min all data lie within 7 the range of the standard deviation. Therefore, a sampling time greater than 20 min would be 8 required to represent the hourly mean. The random data reach a comparable level of data 9 exceeding the hourly mean by 2.80% for averaging over 5min only. Here, sampling for only 10 10 min would be sufficient for representing an hour worth of data." It should be clearly stated that sampling periods of 5-20 minutes would be fine for these specific dataset but that longer sampling periods may be necessary for other environments, especially for measurement sites close to different types of emission sources (e.g. industries).

Fig. 5: Please indicate what "bvf" and "bvfo" mean in the caption.

Fig. 9: As indicated in the main text P18 L19-22, Fig. 9 displays "consecutive 5 min averaging periods within the hour", i.e. 12 independent periods of 5 min. Since the deviation observed depends only on missing VOC variability for the 5-min calculations, shouldn't an average of the 12 residual slopes be zero? It is obviously not zero for each panel of Fig.9.

Fig. 12: Please indicate in the caption what the error bars are.
* * *

---

## Author Response (AR1)

This manuscript reports how OH reactivity values calculated from trace gas measurements could be impacted by unmeasured VOC variability when VOC instruments are not capable of continuous measurements, e.g. Gas Chromatographic instruments. The authors used 1-min continuous measurements of VOCs (PTR-ToFMS) from two different field campaigns to calculate hourly means of OH reactivity. Additional calculations of hourly values were performed by averaging VOCs over different time intervals (5 to 30 min) within each hour to mimic what would have been calculated from GC measurements characterized by various sampling durations. Overall, this study demonstrates that deviations up to 25% could be observed for targeted VOCs. This significant source of errors has to be accounted for when calculated OH reactivity values are compared to direct measurements.

This manuscript is well structured, clear and concise, and will be of interest for the atmospheric community. I therefore recommend publication in ACP after the authors address the following minor comments:

Minor comments:

P2 L12: ":::it sinks are manifold:::" should read ":::OH sinks are manifold:::"

Fixed.

P5 L9: Was the Kore PTR-ToFMS equipped with an ion funnel? It seems so since the study of Barber et al. (2012) is cited. It should be clearly stated in the text and the authors should indicate, if relevant, how the ion funnel impacts the PTRMS response (sensitivity, humidity effects:::).

For ClearfLo the PTR-ToF-MS was not equipped with the ion funnel. Barber et al. (2012) was cited here, because the standard apparatus is described in this paper. The citation has been changed so that it is clearer:

A PTR-ToF-MS (Series I; Kore Technology Ltd., UK) (see standard PTR-MS apparatus in Barber et al. (2012); Thalman et al. (2015))

P5 L13-14: "For background measurements a hydrocarbon trap was employed". Please provide details about this hydrocarbon trap. How efficient was it to scrub hydrocarbons.

How were background measurements performed for OVOCs? How often were the background measurements recorded?

Details were added to the manuscript p.5, l21 - 23:

"For background measurements a hydrocarbon trap (activated carbon filter by Grace Alltech) was employed once during the time period investigated here. Its efficiency was in the range of 87% - 96 %."

P5 L17-18: "The stability of the instrument during the campaign was monitored with a bromobenzene internal standard". Could the authors indicate how stable it was during the campaigns? Was there a need to correct for a drift in sensitivity? If so, how was it done?

During the period analysed here, the instrument was fairly stable and no correction needed to be applied.

Information added, P5, l27:

"Based on these measurements no correction needed to be applied."

P5 L25-26: Was an ozone scrubber used for the GC measurements?

The samples analysed by the instrument passed through around two metres of stainless steel tubing which was heated to 80 °C. This destroys ozone present in the sample.

P5,L26-27, sentence added: " Stainless steel tubing heated to 80°C was used as sampling line destroying ozone present in the sample."

P18 L10-11: "A sampling time of only five minutes can cause a deviation of more than 25%. Accordingly, this would then artificially contribute to missing OH reactivity." This reviewer does not agree with the last sentence, which should be rephrased. The deviation will either lead to a positive or negative bias and will not always appear as missing OH reactivity. This deviation should be discussed as an additional source of errors to account for when measured and calculated OH reactivity values are compared.

The sentence has been rephrased as followed:

Accordingly, this would then artificially contribute to a deviation in OH reactivity, whether it causes a positive or negative bias. Thereby, it is an additional error source when comparing measured total OH reactivity to OH reactivity calculated from GC data.

P18 L21: ": : :(cf. Figure 4): : :" Wrong figure.

Fixed.

P23 L5-10: "As can be seen in Table 9 at 20min still 2.78% of the ClearfLo data exceed their hourly mean. At 30min all data lie within the range of the standard deviation. Therefore, a sampling time greater than 20 min would be required to represent the hourly mean. The random data reach a comparable level of data exceeding the hourly mean by 2.80% for averaging over 5min only. Here, sampling for only 10 min would be sufficient for representing an hour worth of data." It should be clearly stated that sampling periods of 5-20 minutes would be fine for these specific dataset but that longer sampling periods may be necessary for other environments, especially for measurement sites close to different types of emission sources (e.g. industries).

Lines added with regard to the above comment:

The required sampling times mentioned here correspond to the VOC variability of the analysed data sets. Likewise, longer sampling times could be necessary for representing hourly OH reactivity in other environments such as measurements closer to industrial sources. For example, Gilman et al. (2009) have shown, that a much broader range of OH

reactivity of VOC with a high degree in variability can be found in the proximity of heavily industrialised areas like the Houston and Galveston Bay area in Texas, USA.

Fig. 5: Please indicate what "bvf" and "bvfo" mean in the caption.

Done.

Fig. 9: As indicated in the main text P18 L19-22,

Fig. 9 displays "consecutive 5 min averaging periods within the hour", i.e. 12 independent periods of 5 min. Since the deviation observed depends only on missing VOC variability for the 5-min calculations, shouldn't an average of the 12 residual slopes be zero? It is obviously not zero for each panel of Fig.9.

Based on this observation, the different regression models are analysed/compared further on (see Figure 10 and Table 8. On average over all 12 consecutive 5 min intervals, only the linear least square fit and the ratio are close to zero. This is not an issue of the sampling technique as it occurs also in the log-normal randomized data set.

Fig. 12: Please indicate in the caption what the error bars are.

Done. The error bars in Fig. 12 and also in Fig. 13 are the lower and upper limit of the fitted slopes.

The manuscript "What effect does VOC sampling time have on derived OH reactivity?" presented by Sonderfeld et al. is an extended analysis of two different field campaign's high resolution data-sets with regard to the question if and how strong different sampling times and intervals impact the calculation of missing OH reactivity. This question is of relevance for understanding atmospheric photooxidation processes especially when the directly measured total OH reactivity and the budget of individually detected OH sinks such as volatile organic compounds (VOC) are compared. For example, gas chromatographic samples are often collected over a time spanning several minutes whereas the actual analysis of the sample can be as long as one hour. Sonderfeld et al. explore methodically whether a comparison between such low and high resolution data-sets might miss-represent the true variability and average value of the atmospheric observations with respect of the overall OH reactivity. Overall, the data-sets discussed in this manuscript are taken from recent field measurements campaigns, the analysis has been explained carefully, the results are well presented, and the overall conclusions aim to answer the question which was raised within the title. I recommend this paper for publication in ACP and have the following minor specific, general and technical comments.

Reply to specific comments:

SC 1) p.3, l.10- p.4, l.6: The authors present a very good compilation of studies about missing OH reactivity based on the comparison of individually measured OH sinks and the directly detected total OH reactivity. As stated later in the introduction (paragraph p.4, l.14-22) the different time resolutions of OH reactivity and individual compound measurements might bias the comparison hence the resulting missing OH reactivity. With this in mind and following the scope of the entire manuscript, it would be interesting for the reader to get some information about typical instrumentation (e.g. GC) and sampling times (e.g. first 10 min of a 40 min cycle) that have been used for VOC measurements and compared to the directly detected total OH reactivity. Is it possible to include this additional information, either for all or some of the examples that are provided in the text already?

These details can only be extracted from a few of the cited studies. Sampling times are mentioned occasionally, but the analysis process how the different time resolved data are compared to each other is rarely mentioned precisely.

Where possible, information are added to the above mentioned paragraph, now p. 4, L15 – 28.

SC 2) p.3, l.28-29: Indeed both of the studies presented here concluded that the missing OH reactivity is possibly due to unmeasured oxidation products. However, they did not exclude the contribution of undetected forest emissions that may add to the missing reactivity, as well.

Sentence added to p3, l29:
"Undetected biogenic emissions and transport of reactive compounds are also cited as other reasons for missing OH reactivity."

SC 3) p.5, l.14-15 and p.7, l. 9: For both measurement campaigns that are chosen for presenting data in this manuscript a calibration was performed before (and for PARADE also after) the campaign. Was the calibration done at the measurement site or in the laboratory?

The calibration measurements for the ClearfLo data were done in the laboratory. Clarified on p5, l24 now.

The PTR-ToF-MS used for PARADE was calibrated in the laboratory. Mentioned in also on l10, p7 now.

SC 4) Table 2: How did you calculate the accuracy as error for the measurements?

ClearfLo: The accuracy was calculated from the error of the calibration measurements.

PARADE: In Table 2 the given accuracy is the mean uncertainty calculated with error propagation based on uncertainties in calibration, background measurements and fragmentation patterns.

Why did you chose 1 sigma for the ClearfLo campaign and 2.6 sigma for the PARADE campaign as LOD?

The two field campaigns were conducted independently of each other. So, the LOD were calculated in a way to reflect each data set and the format they chose to report these for each of the campaigns. As it does not have an effect on the results of this study, they were not recalculated to match each other, but instead stated clearly.

SC 5) p.11, l.2-3: Since the different monoterpenes have very different reactivities towards OH, it is critical to know their atmospheric composition for calculating the OH reactivity due to monoterpenes. Especially the two contrasting sites presented here might have very different typical ambient monoterpene distributions resulting in different characteristic OH reactivities. Unfortunately, the PTR-MS cannot separate the different monoterpenes and detects all of them as one single signal. Therefore, the authors decided to use the reaction rate of a-pinene only. In case that during the campaigns the monoterpene composition was characterized by accompanying instrumentation (e.g. GC-MS) could you use that data to estimate a typical monoterpene OH reaction rate coefficient? Or in case that you did not have such observations during the campaign, are there any studies in the literature that could give you hints about the typical monoterpene distribution at the campaign sites? A typical monoterpene composition will help you to estimate a typical monoterpene reaction rate with OH that is representative for location and timing of the two campaigns.

That is an important point, when calculating OH reactivity from monoterpene measurements.

For ClearfLo the monoterpene signal of the PTR-ToF-MS was not analysed, so this has no effect on the presented results from ClearfLo.

For PARADE the monoterpene signal was analysed and is presented in this study (Tab 4 and Fig.12). Applying a different reaction rate would affect the calculated OH reactivity in Tab. 4 by shifting them to higher values. Nevertheless, the slope of the correlations analysed here for the monoterpenes does not change, as the reaction rate would be changed in x-axes and y-axes in the same way. A test was done by applying the weighted reaction rate presented in Nölscher et al. (2013), based on GC measurements in spring at the same site. The same slope presented in Fig 12 was observed.

SC 6) p.12, l.19: Why do you need to generate a randomized data set? What do we learn from comparing this fictive distribution of OH reactivities to the field data?

The randomly generated data set was generated and compared to the field data to rule out, that the observed effects arise owing to sampling artefacts. Also it provides a clear counterfactual to the measured data.

SC 7) p.14, l.20-28 and p.16 Table 5: Could you please explain why you chose to not include _R in Table 5?

The mean of ΔR can be expected to be very close to zero and the range will reflect the extreme values, so I found, that this would not add any useful information here. These values could still be added to Tables 5 to 7, if this would be beneficial for the reader.

The standard deviation of _R is used as a measure of variance and presented in Table 5. However, I wonder if it would be more accurate to look at the standard deviation relative to the hourly average _R?

By looking at the residuum ΔR we are investigating the absolute difference/spread between the hourly mean and the shorter sampling time. The relative variance is investigated by the frequency distribution of their ratio and its FWHM.

Also, could you please clarify the physical meaning of the values presented by the Gaussian Fit Centre and the FWHM (full width at half maximum)? You say that the Gaussian function was fitted on the frequency distribution of the ratio of shorter interval averages (R(t<60)) to the hourly average (R(t=60min)). In this case a value of 1 would be calculated for perfect overlap of those two averages and the FWHM would convert to zero, right?

Sentence added now on p15, l 9,10:
"Ideally, the centre of the Gaussian fit is 1, while the full width at half maximum (FWHM) describes the spread of the distribution around its centre."

SC 8) p. 16, l. 7-8: These two sentences seem to have contradicting statements: The small standard deviation of _R highlights the narrow range of calculated OH reactivity. And the high variability of the data is reflected by the relatively high FWHM. Could you please explain? Would you get a different result if looking at the relative standard deviation?

This is directly connected to the reply to SC7). The comparably small standard deviation of ΔR results from the much lower range in OH reactivity of the discussed VOC during PARADE. When looking at the relative spread (FWHM of Gaussian fit to ratio), we find a slightly higher variance compared to ClearfLo.

SC 9) p.18, l.12-15: A more general remark in this context: It could be interesting, regarding the discussion about the difference of variabilities at the two different sites, to have a look at the variability-lifetime relationship as for example presented in Williams et al. 2000 (http://dx.doi.org/10.1029/2000JD900203). Similarly for section 3.4 in which the effect of different VOC classes on OH reactivity is discussed.

Interesting point. Lines added:

"These results are in line with observations from Williams et al. (2000), who investigated the variability-lifetime relationship of VOC measured in an unpolluted region of Surinam based on the standard deviation of the natural logarithm of their concentration. They found a higher variability for toluene compared to acetone and methanol. Compounds with a lifetime below 2 days did not seem to fit into this relationship."

SC 10) p. 26, Figure 13: The effect of VOC variability on the calculated OH reactivity is presented in this figure for the ClearfLo campaign. How does it look like for the PARADE campaign data? Are there significant differences?

The share of OH reactivity based on the different classes of VOC was calculated from the GC measurements, which provides a wide range of VOC, that were done during ClearfLo at the same site like the PTR-ToF-MS measurements. GC data from the PARADE campaign were not analysed in this study.

SC 11) p. 28, l. 4-5: The missing variability in VOC data, that you mention here, is only due to the short interval sampling time. Is this correct?

Yes. Changed to missed.

SC 12) p. 28, l.5-6: The divergence between 5 min and 60 min averaged calculated OH reactivity is given here to be between 1-28% (PARADE) and 0-44% (ClearfLo). These numbers appear in the text for the first time at this point within the conclusions. Could you please include some reference in the text beforehand? And also it would be good to stronger point out the conditions and the statistical test (e.g. first 5 min of hour, consecutive 5 min intervals, regression methods, number of data points, effect of different VOC classes, :  :  :) that lead to the greatest divergence.

Thanks for pointing this out. Actually for PARADE the wrong range was given, which should have been 2 - 26% and is now corrected

Line added to P20:

"Depending on the selected 5 min interval the bvf resulted in a divergence of - 0.1% to 44 % for ClearfLo, 1% to 13% for PAR1, - 3% to 26% for PAR2 and – 2% to 10%  for the randomised data."

Reply to general comments:

GC 1) Within the presented study you solely compare VOC data with the OH reactivity calculation based on the measurement of individual OH sink compounds. Do you have any directly measured total OH reactivity data available to compare to?

Direct OH reactivity measurements as part of ClearfLo were made during the summer IOP (22 July to 18 August 2012) (Whalley et al., 2016) and can thereby not directly be compared to the here presented PTR-TOF-MS data set from the winter IOP.

During PARADE OH reactivity was directly measured from a branch enclosure system. As they focus on the biogenic emissions of a single tree, they are not directly representative to the VOC mixture observed at the top of Kleiner Feldberg and were not added to this study.

GC 2) For the statistical analysis of the two field campaigns (ClearfLo and PARADE) the entire data-set was used. How does your overall conclusion depend on the time of the day?

Did you test the small sampling interval averages against the hourly averaging for example for day and nighttime data only? Is it possible that some VOC show decreased variability within the nocturnal boundary layer whereas during daytime the close distance to emission sources and turbulent mixing increase their overall variability?

This is an interesting aspect, which would be worth exploring in more detail. However, the analysed data sets only cover a short amount of time (ClearfLo – 1 week; PARADE – 2 times 1 week), which seems not to be sufficient for good statistics over a diurnal cycle. No day-/nighttime effects were analysed here. The PARADE campaign provides a data set of 4 weeks in total and diurnal cycles were observed for some VOC. The complete analysis described in the manuscript would need to be repeated to analyse day/night effects.

Reply to technical comments:

TC 1) p.2, l.10: "Its actual concentration being determined by the balance between its sources and sinks." It seems to me that the verb in this sentence needs to be "is" rather than "being".

Changed.

TC 2) p.2, l.17: Here, a list of references about in-situ measurements of OH reactivity is provided. However, it should be indicated (e.g. with "e.g.") that this list only presents a fraction of the actual literature.

Changed.

TC 3) p.3, l. 10: ": : : good agreement between measured and calculated OH reactivity have been found." It should be "has" instead of "have".

Fixed.

TC 4) Table 1: Are here averages or median values presented for the mixing ratios and concentrations? What is the given uncertainty? Standard deviation?

The table reports mean and stdev of the mixing ratio and concentration, the OH reactivity is calculated from that mean value. The table caption is has been changed to make that clear.

TC 5) p.7, l.25: I think you do not need the "whether" in this sentence.

Right.

TC 6) p.8, l.8: Here you repeat yourself by having "values" twice in one sentence.

Changed.

TC 7) p.10 Figure 2: It would be great, if you could add a legend to the two graphs explaining the different markers used.

Added.

TC 8) p.10 Table 3: Table 3 basically repeats what is shown in Figure 2. I wonder if it is really necessary to include the same information twice. You might want to decide whether to present the figure or the table. Also, what units did you use to present the range of VOC mixing rations in Table 3?

Units added to the table's caption.

TC 9) p.11 Table 4: Please correct the format of the VOC reactivity unit. Also, to be precise it is the OH reactivity due to the selected VOC.

Corrected.

TC 10) p.12, section heading and terminology: The term "VOC reactivity" can be misleading because atmospheric VOC typically react with various oxidants such as $O_3$, $NO_3$, Cl or OH. Hence, it is more precise to use the terminology OH reactivity. This applies already for most of the presented manuscript (e.g. Title, Figure 3 ect.) but should be checked for consistency, especially in this section 2.2.

The term "VOC reactivity" has been changed to "OH reactivity (of VOC)" throughout the manuscript.

TC 11) p.12, l.13: In the previous section you define the notation for different OH reactivity calculations which depends on the instrumentation, campaign and compounds taken into account. The example shows that OVOC during the ClearfLo campaign only includes acetone. Probably during the Parade campaign it would also include methanol. Then in section 2.2 the OH reactivity was calculated for VOC detected by GC during the ClearfLo campaign. However, what do you mean with TVOC as referred to in line 13?

Explained in line 6 on page 12 now.

TC 12) p.18, l. 20-21: Here, it is referred to Figure 4, which shows the randomly generated data set. In the context of presenting residual slopes as in Figure 9, I found this confusing and it might be a mistake.

It should be Fig. 5. Corrected.

Also, from this point on you look at the residual slopes (as plotted in Figure 9). In the figures and sections before (e.g. Fig. 6, 7, 8) the slope was shown. Is there a reason for not being consistent about that?

The residual of the slopes was introduced to highlight the deviation from the ideal slope of one.

TC 13) p. 24, Table 9: With "Random numbers" do you mean the "Randomly generated data set"?

Yes.

[revised manuscript text omitted]

$\cancel{\mu = \log\left(\dfrac{m}{\sqrt{1+\dfrac{sd^2}{m^2}}}\right)}\ \mu = \log\left(\dfrac{m}{\sqrt{1+\dfrac{sd^2}{m^2}}}\right)$

(2)

$\cancel{\sigma = \sqrt{\log\left(1+\dfrac{sd^2}{m^2}\right)}}\ \sigma = \sqrt{\log\left(1+\dfrac{sd^2}{m^2}\right)}$

(3)

A log-normal distribution of a total of 8040 random numbers was generated using the dlnorm (#, μ, σ)-function in R. This provides a set of data comparable to 134 hours of OH reactivity measurements with a time resolution of 1 min. Figure 4 shows the random data set as a time series together with the hourly mean containing 60 data points. On observation of Figure 4, it becomes obvious that the range of the hourly average is very small with a standard deviation of $0.034 \text{ s}^{-1}$.

[Figure]

[Figure]

Figure 4: Time series of randomly generated log-normal data set containing 8040 numbers. The "1min" data are shown in black. The average over 60 data points is plotted in red.

**3    Results and Discussion**

Initially the OH reactivity $\cancel{R_{PTR}^{VOC}}$ $R_{PTR}^{VOC}$ was calculated from the PTR data for ClearfLo and PARADE. For both campaigns the signals of acetone, toluene and xylene (referred to generally as VOC) were used. The effects of differing sampling intervals on the derived reactivity were explored. For each campaign and VOC dataset a correlation of the average values of $R_{PTR}^{VOC,t}$ for different intervals (t = 5, 10, 20 and 30 min) against the 60 min average $R_{PTR}^{VOC,60}$ was calculated. The intervals are chosen to be the first t minutes of each hour to simulate the initiation of a GC sequence, thus the 10 min average also covers the 5 min averaging period and so on.

Figure 5 shows the linear correlation of the 5 min average $R_{PTR,CL}^{VOC,5}$ versus the 60 min value $R_{PTR,CL}^{VOC,60}$ for the ClearfLo winter campaign. Data were fitted with a bivariate regression line with an intercept (bvf) and forced through the origin (bvfo). The deviation from the slope of the linear regression to a unity gradient $m_{res} = \left( m_{R^{<60}/R^{60}} - 1 \right)$ is taken as a measure of how well the value of hourly OH reactivity is represented by the shorter interval average and is further referred to as the residual slope.

**3.1  Effects of different sampling intervals**

The slopes of both fits in Figure 5 are below 1.0, indicating an under prediction of the reactivity during ClearfLo by the value calculated from the first 5 min of each hour. In this case there is only a small deviation (1.3%) from a unity gradient (see Figure 6). For all averaging intervals the slope is equal to 1 in the range of the uncertainties of the fit.

**3.2**

For the different averaging intervals the difference to the hourly average ( $\Delta R = R_{PTR,CL}^{VOC,t<60} - R_{PTR,CL}^{VOC,t=60}$ ) was calculated and their standard deviations are given in Table 5 as a measure of variance. $\Delta R$ generally decreases with increasing averaging time. Also presented in Table 5 are the results from fitted Gaussian functions to the frequency distribution of the ratio of the shorter interval averages to the 60 min average. Bins of 0.1 were chosen for the frequency distributions. Ideally, the centre of the Gaussian fit is 1, while the full width at half maximum (FWHM) describes the spread of the distribution around its centre. The standard deviation of $\Delta R$ as well as the (FWHM decrease, when averages are calculated for longer intervals. The centre of all Gaussian fits achieve 0.99.

[Figure]

[Figure]

————————————————————————

Figure 5: Linear correlation with bivariate fit (bvf: fit with intercept, bvfo: fit forced through the origin) of the OH reactivity calculated from the signals of acetone, toluene and xylene– for average intervals of 5 min and 60 min for ClearfLo. The standard deviation of the 5 min means are plotted as error bars.

[Figure]

[Figure]

Figure 6: Development of the slope of the correlation of OH reactivity $R_{PTR,CL}^{VOC,t}$ $R_{PTR,CL}^{VOC,t}$

depending on the sampling interval for ClearfLo. Slopes for bvfo (black) and bvf (blue) with their lower and upper limits are shown.

| Notation | Time interval | $\Delta R$ Stdev | Gaussian Fit Centre | FWHM |
|----------|---------------|---------|--------|------|

| | (min) | (s⁻¹) | | |
|---|---|---|---|---|
| $R_{PTR,CL}^{VOC,5}$ | 5 | 0.12 | 0.998 ± 0.011 | 0.337 ± 0.025 |
| $R_{PTR,CL}^{VOC,10}$ | 10 | 0.12 | 0.997 ± 0.008 | 0.244 ± 0.020 |
| $R_{PTR,CL}^{VOC,20}$ | 20 | 0.10 | 0.988 ± 0.006 | 0.246 ± 0.013 |
| $R_{PTR,CL}^{VOC,30}$ | 30 | 0.06 | 0.992 ± 0.004 | 0.198 ± 0.009 |

For Period 1 of the PARADE data (PAR1) the results show a slope greater than 1 (Figure 7). The high variability of the data is reflected by a higher divergence of the slopes of 1.13 for bvf fit and 1.05 for the bvfo fit based on 5 min averaged data. The small standard deviations of $\Delta R$ given in Table 6 highlight the narrow range of calculated OH reactivity $R_{PTR,PAR1}^{VOC}$ However, the high variability of the data is reflected by the FWHM of the frequency distributions of the ratios which is higher for each interval when compared to ClearfLo.

[Figure]

[Figure]

Figure 7: Development of the slope of the correlation of OH reactivity $\cancel{R^{VOC,t}_{PTR,PAR1}}$ $R^{VOC,t}_{PTR,PAR1}$

depending on the averaging time for PARADE - Period 1.

$\cancel{\Delta R}$

| Notation | Time interval |  Stdev | Gaussian Fit Center | FWHM |
|---|---|---|---|---|

| | (min) | ($s^{-1}$) | | |
|---|---|---|---|---|
| $R_{PTR,PAR1}^{VOC,5}$ | 5 | 0.016 | 0.980 ± 0.011 | 0.379 ± 0.027 |
| $R_{PTR,PAR1}^{VOC,10}$ | 10 | 0.015 | 0.976 ± 0.012 | 0.353 ± 0.026 |
| $R_{PTR,PAR1}^{VOC,20}$ | 20 | 0.012 | 0.997 ± 0.013 | 0.310 ± 0.030 |
| $R_{PTR,PAR1}^{VOC,30}$ | 30 | 0.008 | 0.995 ± 0.009 | 0.273 ± 0.020 |

For Period 2 (PAR2), an over prediction of the OH reactivity $R_{PTR,PAR2}^{VOC}$ $R_{PTR,PAR2}^{VOC}$ can be observed again (Figure 8), but with an even greater slope of 1.26. In both periods of PARADE the slope approaches a value of 1 as increasing averaging time is taking more of the variability within one hour into account. Standard deviations of $\Delta R$ $\Delta R$ and FWHM values are similar to Period 1 of the PARADE data, while the centres of the Gaussians are closer to 1 (Table 7).

[Figure]

[Figure]

Figure 8: Development of the slope of the correlation of OH reactivity $\cancel{R_{PTR,PAR2}^{VOC,t}}$ $\underline{R_{PTR,PAR2}^{VOC,t}}$

depending on the averaging time for PARADE - Period 2.

$\cancel{\Delta R}$

$\cancel{R_{PTR,PAR2}^{VOC,t<60} / R_{PTR,PAR2}^{VOC,t=60}}$

[revised manuscript text omitted]